# Inadequate Regulation of the Geological Aspects of Shale Exploitation in the UK

**DOI:** 10.3390/ijerph17196946

**Published:** 2020-09-23

**Authors:** David K. Smythe

**Affiliations:** College of Science and Engineering, University of Glasgow, Glasgow G12 8QQ, UK; david.smythe@glasgow.ac.uk

**Keywords:** fracking, unconventional, hydrocarbon, exploration, drilling, well, regulation

## Abstract

Unconventional oil and gas exploitation, which has developed in the UK since 2009, is regulated by four main agencies: The Oil and Gas Authority, the Environment Agency, the Health and Safety Executive and local Mineral Planning Authorities (usually county councils). The British Geological Survey only has an advisory role, as have ad hoc expert committees. I firstly define terms, and summarise the remits of the regulators and background history. Fourteen case histories are then discussed, comprising most of the unconventional exploitation to date; these cases demonstrate the failure of regulation of the geological aspects of fracking operations in the UK. The regulators let inadequacies in geological understanding, and even mendacious geological interpretations by the hydrocarbon operators slip through the net. There are potentially severe implications for environmental safety—if and when permits are granted. Geological pathways, if not properly understood and mitigated, may lead to long-term pollution of groundwater and surface water; methane and H_2_S emissions. Induced earthquakes have not been well regulated. The case histories demonstrate a laissez-faire and frequently incompetent regulatory regime, devised for the pre-unconventional era, and which has no geological oversight or insight.

## 1. Introduction

During the last decade, the era of fracking, the UK regulatory system for onshore hydrocarbons [1] has been vaunted by the government since the inception of fracking as “robust” [2] and that it: “should be the gold standard that others should aspire to” [3].

This is not and has never been the case [4,5]. The regulation was designed for the pre-fracking era; “old regulations are being used to govern a new technology” [6]. In practice, as the case histories below will show, the regulation has been wanting. In particular, the responsibility for scrutinising the geological aspects of unconventional exploration tends to fall between the stools of the several regulators.

The most important regulator from the geological point of view would appear to be the Oil and Gas Authority (OGA). Its role is “to regulate, influence and promote the UK oil and gas industry in order to maximise the economic recovery of the UK’s oil and gas resources”. However, as we shall see, the Environment Agency (and its equivalent in Scotland, the Scottish Environmental Protection Agency) turns out to have the main responsibility concerning geology.

Scrutiny of the geology, as presented by operators during the planning process, is crucial because of the potential pollution which may occur, and which may be an order of magnitude larger than that from conventional oil and gas activities. This includes groundwater and surface water pollution by brine, heavy metals, radioactive elements (both heavy and light) and by extremely toxic chemicals used in fracking like polyacrylamide; methane and H_2_S emissions; disposal of produced water (a by-product of the fracking process); and earthquakes.

The two major aquifers in the UK are the Chalk and the Sherwood Sandstone Group. Each of these usually lies above, or adjacent to, formations that are being exploited unconventionally—the Kimmeridge Clay Formation and the Bowland Shale, respectively [7]. Faulting, which is complex in the UK, compared to, say the shale basins of the USA, provides the pathway between deep and shallow groundwater systems. Therefore, the intrinsic risk of contamination of the aquifers is high.

Firstly, I summarise some necessary definitions of terms used in unconventional exploitation, for the benefit of non-technical readers. Then I review and analyse the geological aspects of the regulators, highlighting their behaviour and performance in the face of the new problems created by unconventional exploration. Such an emphasis on geology has not hitherto been presented. A series of fourteen case histories follows. The technical weaknesses or failures of the industry are pointed out where necessary to highlight how and why regulation has either failed, or is inadequate, in each case. The review is not a critique of the industry per se, but of its regulation.

These geological case histories include Scottish examples of coal bed methane (CBM) exploration proposals, and some English examples include cases in which the exploration is presented by the industry as conventional, but in which an agenda of potential future unconventional activity is implicit. They are presented in approximately chronological order, grouped by region, as follows, with the number of separate case histories shown as follows:The Weald: 2008 to 2014    2Scotland: coal bed methane   2Lancashire: 2009–2014     1The Weald: 2014 to date    4Lancashire: 2014 to date    2East Midlands        1Cheshire           1South-eastern England    1

The primarily chronological order is necessary to trace the evolution of regulation over the last decade. Extra background information is provided in a single Appendix A document, divided under different headings.

## 2. Onshore Licensing and Regulation—General

### 2.1. Licensing

The main licensing regime for onshore oil and gas exploration and production in the UK dates back to the Petroleum (Production) Act 1934 [8], even though some licences had been issued under a 1918 Act [9]. The current regime is controlled by the Petroleum Act 1998 [10]. This date precedes the advent of ‘unconventional’ hydrocarbon exploration (terms are defined below), which came of age in the USA around 2005. The 1998 Act has since been modified to incorporate the new technique of hydraulic fracturing (‘fracking’), but definitions used by the UK regulators are at times vague, do not correspond to scientific definitions [11], and can be internally inconsistent [12].

Blocks of land for licensing are offered by the government (currently the Department for Business, Energy and Industrial Strategy, DBEIS) in periodic licence rounds. Applicants may be awarded one or more blocks at the discretion of the Secretary of State. If there is competition for the same block, the ‘best’ applicant will be awarded a Petroleum and Exploration Development Licence (PEDL). Inadequate applications may result in a particular block not being licensed. The system is thus discretionary; the fees paid for acquiring a licence block are minimal.

The Department of Energy controlled the licensing from 1973 to 1992, at which point its core responsibilities, including licensing, were transferred back to the Department of Trade and Industry (DTI). Responsibility then passed to the new Department of Energy and Climate Change (DECC) from 2008 to 2016, following which DECC was absorbed into DBEIS. The Oil and Gas Authority (OGA) had previously been established in April 2015 as an executive agency of DBEIS, then on 1 October 2016 it was incorporated as a Government Company, with the Secretary of State as the sole shareholder. Currently the OGA has the responsibility for licensing.

Recently, legal doubt has arisen about whether a licence is administered by predetermined regulation, or is essentially a contract between two parties [13] and therefore open to variation by mutual agreement. Such doubt has arisen due to the behaviour of the OGA regarding the grant of extensions to the duration of some onshore licences.

### 2.2. Regulation

Regulation of the onshore hydrocarbon exploration and production industry is shared between four separate entities [14]:OGA; it conducts licensing rounds and awards blocks (see above), also monitors subsequent activity.Environment Agency (EA; the equivalent body in Scotland is the Scottish Environmental Protection Agency, SEPA); it issues a permit when it is satisfied that the proposed activity will not cause environmental harm.Minerals Planning Authority (MPA); the MPA is the County Council (in two-tier parts of the country), or else the Unitary Authority, or else the National Park Authority; it grants planning consent [15].Health and Safety Executive (HSE); it issues a drilling permit once it is satisfied that the well design is safe; subsequently it visits and monitors the drilling and production activity.

How these separate entities interact is discussed in the next section. In many ways the MPA is central to the whole process, as shown by an organogram from 2013 (Appendix A).

From time to time expert committees have been set up and public consultations (in addition to the routine local authority planning application consultations) have been held. These include the joint Royal Society and Royal Academy of Engineering working group, the Scottish expert committee, and the ad hoc Newdigate earthquake working group. Public consultations are routine practice for planning applications to MPAs; some of them attracted record numbers of submissions because of the controversial nature of fracking. A national Scottish consultation on unconventionals was held in 2015.

In two of the case histories discussed below the development proposals were sent to a public inquiry. In each case the inquiry report was called in for determination by the relevant minister, but, at the time of writing, no determination has been handed down in either case.

## 3. Scrutiny of the Geological Aspects of the Licensing System

### 3.1. Definitions

Prior to 2010 all exploration in the UK was ostensibly conventional, but the dividing line between conventional and unconventional became blurred by Cuadrilla’s planning application of January 2010 to drill an exploratory well at Balcombe, Sussex. This case history is discussed below.

A *conventional* oil or gas resource is hosted in a well-defined finite volume of rock called the reservoir; the resource can be extracted without needing to alter the bulk rock properties. The oil or gas will have been formed elsewhere in a source rock, and will have migrated over geological time to become trapped in the reservoir. Figure 1 shows a cartoon cross-section of oil and gas collecting in geological structures, inhibited from percolating upwards by a seal. The oil or gas, being less dense than the brine it displaces, tends to flow upwards under gravity.

The trapped oil or gas may be recovered to the surface by natural flow pressure up a borehole; however, steam or water may be injected to direct or encourage the flow. The wellbore and its immediate vicinity (<1 m radius) may also be cleaned up, or ‘washed’, to remove mud or limescale.

In contrast, an operational or economic definition [16] of an *unconventional* resource is:

“one that cannot be produced at economic flow rates or that does not produce economic volumes of oil and gas without the assistance of massive stimulation treatments or special recovery processes and technologies”.

The problem with the latter part of such a definition is that it evolves with time. A resource that may be considered unconventional now may not be so in the future, as leading-edge technology evolves and becomes routine. The US Geological Survey does not explicitly define “unconventional”; its publications prefer to describe “continuous petroleum accumulations”.

Here we define unconventional exploration as including all techniques which permanently increase the permeability of shale, converting it from a source rock to a reservoir rock. *Permeability* is a measure of how easily a fluid flows through a medium when subjected to a pressure gradient. The principal and best-known method of unconventional extraction of gas or oil from shale is *hydraulic fracturing*, now referred to simply as *fracking*. *Coal bed methane extraction* is also an unconventional method because the coal seams need to be fracked to release the methane. In vulgar parlance, ‘fracking’ has evolved to embrace all unconventional extraction methods, including acidisation [12], even though some of these methods may not employ high pressure hydraulic fracturing of shale rock. Although the industry itself invented the neologism ‘fracking’, it now tries to avoid using the term because of its pejorative implications [17].

The generally accepted boundary between ‘tight’ reservoir rocks requiring massive (unconventional) treatment and conventional reservoir rocks is a permeability of 0.1 millidarcies (Appendix A).

### 3.2. Pre-Fracking Practice in the UK

Up to 2010, all drilling for oil and gas in the onshore UK was conventional. Between about 2007 and 2015 applications for coal bed methane exploration, which is a form of unconventional activity, were made, but none of these resulted in licensing and drilling.

A hydrocarbon company wishing to acquire a PEDL, and thereby become an *operator* (the single responsible legal entity, perhaps in financial partnership with other companies) would first study the geology in the area of interest, and identify specific *leads*. These are defined volumes of the subsurface which may contain oil or gas. If a risk assessment is carried out, a lead may be upgraded to a *prospect*; that is, a specific target ready to be drilled. This prior study is not carried out in vacuo; one of the merits of the UK licensing system is the preservation, archiving and eventual release of all commercially-obtained exploration drilling and seismic reflection data. In the sedimentary basins where hydrocarbons occur this vast database greatly outweighs the subsurface and surface mapping studies of the British Geological Survey (BGS), the oldest national geological survey in the world, established in 1835.

The confidential well logs and seismic data acquired commercially in previous exploration campaigns are released after a confidentiality period of five years. The seismic data are available for online inspection at the UK Onshore Geophysical Library [18] (UKOGL), which also provides the locations of the wells and their *tops* (depths to the top of each relevant geological layer) if the well has been released; the well logs themselves may be purchased from other OGA agents. The cost of obtaining all the relevant data for a typical block (a 20 km × 20 km square, based on the National Grid), may amount to a few thousand pounds. This is a tiny fraction of the cost of drilling an onshore well (GBP ~1–2 M) or acquiring new seismic data (GBP ~10 K/linear kilometre).

Licences are awarded on a discretionary work programme-based bid basis. In the UK sector there have only ever been three sealed cash bid auctions [19], which were all for offshore blocks in 1971–2, 1982–3 and 1984–5. I have personal experience of how the PEDL licence application system works, having sat on the government side on one occasion with Department of Energy officials interviewing BP in 1984, and on the other side of the table at a similar DECC interview in 2009, when I represented a client who was applying for onshore licences in the south of England. I am probably the only person who has had such an experience of the interview process at work from both sides.

At the application interview the applicant presents his or her ideas and work plans for the leads and prospects identified, but in my experience the geological interpretation carries little weight, per se. What counts most is the proposed work programme, which is assessed by a point-scoring system, weighing up, for example:How many firm wells will be drilled;How many conditional wells will be drilled (depending on prior studies);Reprocessing of existing seismic data;How many linear or square kilometres of new 2D or 3D seismic data, respectively, will be acquired;Any other data acquisition methods, e.g., reprocessing of existing data, or novel analysis tools, to be used.

In short, the regulator (currently the OGA) takes little or no account of the geological interpretation that is proposed. As we shall see from the case histories, this did not matter when the exploration was conventional, but with unconventional resource exploration it has become crucial.

Once a PEDL is awarded, the new operator must fulfil the conditions promised in the work programme within the Initial Term of the licence agreement. This is normally of five years duration, and the operator also thereby agrees that the data will become public, even if the PEDL is retained after the period of confidentiality has expired. Background checks on the financial viability of the operator are also made.

Such an award system used to be robust. There was a boom in onshore exploration activity in the 1970s and into the 1980s, up to early 1986, when the price of Brent crude (the main European measure of oil trading) fell from USD 29 per barrel to USD 13 [20]. This price collapse put an end to most UK onshore oil exploration. Up till then the bulk of the exploration had been carried out by the oil majors—BP, Shell, Conoco, British Gas Corporation (BGC), etc.—with only a handful of minor independent (stock market unlisted, or private equity) companies playing a part.

The technical expertise and financial probity of these companies was not in question. In addition, the environmental risks of drilling vertical wells are small. If a wildcat exploration well came in ‘dry’, because the prognosis turned out to be incorrect, the only loss was to the investors. The well would be plugged and abandoned, and there will be little or no lasting damage to the environment.

During the 1980s ‘extended-reach’ horizontal drilling techniques were developed. In the onshore UK domain, this new method was pioneered by BP, which had previously discovered the Wytch Farm field in Dorset in 1974 in partnership with BGC. These wells enabled the field, which is the largest onshore oilfield in western Europe, to produce oil from some 12 km out into Bournemouth Bay from a single drilling pad onshore south of Poole Harbour [21]. It should be noted that the exemplary practice by the operator at Poole Harbour applies to horizontal drilling through the sandstone reservoir, but has little in common with the later development of horizontal fracked wells in shale. Fracking has never been carried out at Wytch Farm, contrary to some opinions [22] which try to cite Wytch Farm (Appendix A) as an example of environmentally safe practice, applicable to fracking of shale wells.

### 3.3. The Advent of Fracking

The shale gas boom in the USA started in about 2006, and reached a peak in 2011. It resulted from the marriage of two technologies; the development of extended-reach drilling mentioned above, together with the successful application of hydraulic fracturing (fracking) of shale to increase its permeability. The latter technique came of age in about 2003. By 2008 the apparent success of the unconventional shale gas industry in the USA was beginning to excite both exploration companies and the government in the UK.

An important difference between the geology of shale basins in the USA and those of Europe is that the former are geologically very simple, with thin shale layers and largely no faulting (vertical and/or horizontal offsets of the layers). This is shown in Figure 1a. However, in Europe the shale is found in faulted basins where the shale is much thicker. Such faults may act as conduits or barriers to the flow of groundwater [23], as illustrated in Figure 1b, and also act as potential conduits for contamination from a fracked shale to near-surface groundwater and to the atmosphere. This danger, which was mentioned in an early US oil industry review [24] referring to shallow fault-bounded accumulations in which the fault “penetrates to or near the surface” was considered to be rare, but in Europe such faulting, linking deep structures to the surface, is commonplace. Faulting was recognised as a potential environmental hazard, if fracking were to be performed, in reports produced for governments in Germany [25] and in France [26]. These reports (and others) helped to lead to moratoria on fracking in these two countries.

The water industry is aware of faults as conduits, and, indeed, takes advantage of that fact. For example, the United Utilities Cumbrian website in 2011 stated (apropos of the Sherwood Sandstone Group at outcrop):

“03 February 2011: We are using drilling rigs to explore for one of Cumbria’s most precious natural resources—water. Our specialist teams have plunged four boreholes up to 120 m deep in fields south of Egremont to pave the way for a potential new groundwater supply. Project manager Danny Brennan said: ‘*The boreholes have been sited to **target geological faults to give the best access to the yields***’” (emphasis added).

The report is currently still available on the website WaterBriefing [27]. The Sherwood Sandstone Group (SSG) is one of the two most important aquifers in the UK, the other being the Chalk. Recently, Medici et al. [28] used deep boreholes from Sellafield, West Cumbria, to demonstrate that extensional faults at c. 800–900 m below sea level within the SSG are preferential flow pathways. More generally, normal faults in fluvial siliciclastic (quartz-bearing sedimentary) deposits, such as at Sellafield, represent flow heterogeneities of tectonic origin which typically act as favourable flow pathways, but normal faults in aeolian deposits are commonly associated with high-density networks of deformation bands; as such, these faults act to reduce well-test-scale hydraulic conductivity [29]. So, the environmental origin of the siliciclastic SSG is crucial in determining whether faults in the SSG act as conduits or as barriers to fluid flow.

In contrast, the BGS issued two reports [30,31] written by hydrogeologists at the same epoch, about the contamination risks of shale gas exploitation, but used inappropriate simple US geological analogies, and did not even mention the word ‘fault’. The latter report is discussed in Section 3.9 below.

It is now possible to directly image the passage of fluids up faults. A good-quality 3D seismic volume is required; this implies that only marine seismic data may be good enough, not onshore data. Special processing techniques are applied to visualise the gas flow (see the extract from a recent monograph [32], reproduced in Appendix A: Aminzadeh et al., 2013 extract). The lesson from such studies is that the subsurface is always somewhat ‘leaky’, and that fluid flow in sedimentary basins is a dynamic, ongoing process.

### 3.4. The Oil and Gas Authority

The OGA is a limited company, with the Secretary of State for Business, Energy and Industrial Strategy as the sole shareholder. Its role as the regulator of the UK oil and gas industry means that it carries out functions of public administration for the purposes of regulation under the Environmental Information Regulations 2004.

The OGA became an Executive Agency on 1 April 2015, thus creating operational independence from DECC (now the Department for Business, Energy and Industrial Strategy) to the fullest extent possible within the established boundaries. That gave it direct responsibility for exploration and development decisions and approvals. Its stated ambition is to be “a world-leading authority setting the framework for a sustainable and competitive UK oil and gas industry”, whose purpose is “to maximise the economic recovery of oil and gas”. Although the website pays lip service to “supporting the energy transition to a low carbon economy”, the whole thrust of the OGA appears to be maintaining and enhancing a ‘business as usual’ approach to the UK’s hydrocarbon sector, as if the climate crisis never existed.

Most state regimes controlling their oil and gas resources abhor licence speculation; that is, the acquisition of licences which are then held either without development, akin to land-banking, and/or are traded with other companies (see Appendix A: A note on licensing regimes outside of the UK). Historically, the UK system described above prevented such unproductive activity; however, under the OGA this protection has been loosened. Despite the OGA’s assurance that by splitting the licence term into three parts:

“It allows the OGA to ensure that licensees do not retain valuable exclusivity of hydrocarbon exploration and extraction without doing enough work for this to be justified”.[33]

In practice, however, licences are now awarded when:There is little realistic prospect of the operator fulfilling the licence obligations;The licence obligations themselves may be derisory;Extensions to the licensing terms are granted with little or no scrutiny or justification;The financial probity of the operator is in doubt.

Examples of the above failings 1–3 are given in the case history section below. Many examples of no. 4, which is beyond the scope of this review, may be found on the website Drill or Drop [34]. 

The legal status of the licence itself seems to be changing, from that of a piece of legislation enshrined in law, to a that of a private contract between the licensee (the operator) and the OGA [13].

The hands-off approach to licensing by the OGA extends to the actual exploration and drilling phase, which is left to the Mineral Planning Authority to regulate, and which in turn leans on the EA. However, one aspect of geological importance in which the OGA plays an ongoing role is in the case of induced seismic activity. Three case histories (Lancashire Preese Hall-1, Lancashire Preston New Road, and Weald Horse Hill) concern such seismicity. The seismicity induced by fracking at Preese Hall-1, Lancashire, in 2011 came as a surprise, because the UK is perceived to have only moderate seismicity, based on good historical documentation going back for a millennium [35]. As a result, the expert review of fracking for shale gas by two UK academic societies set up the following year (discussed in Section 3.9 below) placed undue emphasis on induced seismicity, perhaps to the detriment of other valid concerns. The local magnitudes M_L_ of the largest induced earthquakes to date are M_L_ = 2.3 (Preese Hall), 1.5 (Preston New Road), and 3.0 (near Horse Hill, Surrey). To put these figures in context, one natural (tectonic) earthquake of magnitude 3.0 occurs in the UK every 3–4 years [35].

There were no PEDLs awarded in Scotland in the 14th round of licensing, where a moratorium against fracking was imposed by the Scottish government in January 2015 and upheld by MSPs in October 2017.

Apart from seismicity issues, the OGA thus explicitly avoids getting involved in questions of geology, as was made clear by a letter from its predecessor DECC to Lancashire County Council, the local MPA, in 2014 (Appendix A: Comments on letter from DECC to LCC). It stated that such questions were the responsibility of the EA.

### 3.5. Mineral Planning Authority

The mineral planning authority (MPA) is usually a local authority such as a county council. Appendix A shows an organogram of the central role played by the MPA. An operator will submit a planning application to the MPA, normally accompanied by an environmental statement and much other documentation. In addition to details of the site to be developed, the operator may provide discretionary information on the background history of the exploration, the geology, the drilling methods and equipment to be used, and so on, in addition to the type of information that the MPA planners will be more familiar with, such as disturbance, traffic movements, adverse effects on wildlife, etc.

The MPA does not require that the Environment Agency (SEPA in Scotland) and the Health and Safety Executive (HSE) issue their respective permits before granting planning approval; however the MPA must confirm that all relevant planning conditions (including permits from the statutory consultees EA and HSE) have been discharged before consent to drill is issued by the OGA [1]. The organogram (Appendix A) and the more complex ‘roadmap’ flow chart [1] show that there is a back-and-forth process of consent granting which can lead to confusion. The Brockham case history (discussed in Section 4.7.2 and Section 5.7.3 below) is an example of such confusion.

The MPA decision on whether or not to grant permission is normally taken by its planning committee (comprising councillors), which in turn relies on an Officer Report. The Officer Report is prepared by professional employees of the MPA. Any public consultation submissions should be taken into account in recommendations made to the planning committee by the Officer Report. The planning committee often reaches its decision in public.

The MPA and its employees do not have any special expertise in geology. There is therefore a risk that its planning decision may be overly dependent on what the developer chooses to disclose about the geology. An example of this is the common statement in mineral plans, directly quoting the planning guidance: “Minerals can only be worked where they occur”, so that if a prospective operator asserts that such-and-such a place is the only location suitable for drilling, then the MPA is likely to accept that assertion without question. This localisation argument is usually valid where the target is conventional, but carries little weight in the case of unconventional exploitation of shale. In the former case the target area may typically be less than 10 km^2^, whereas in the shale case a whole licence block of 400 km^2^ may be equally exploitable. Examples are provided below.

The MPA is constrained in its freedom of judgment by national guidance. Here is an example of a Joint Minerals Local Plan statement by West Sussex County Council [36] on hydrocarbon development (jointly with the South Downs National Park Authority):

“Planning permission is only one stage in the process of securing consent to drill. The Authorities must assume that the other regulatory bodies (the Environment Agency, Health and Safety Executive and Oil and Gas Authority) operate as intended. However, consulting with the other regulatory bodies on planning applications helps to ensure that the Authorities can be satisfied that the issues they cover can and will be adequately addressed. National guidance is very clear that issues covered by other regulators including emissions, well and surface equipment integrity, processes controlling drilling and extraction, and health and safety should not be addressed by the planning process”.(citations omitted)

An MPA may occasionally seek outside expert advice on geological questions. This is right and proper, although the choice of expert(s) to be consulted for their advice may in itself be questionable. Examples are given below in which an expert was poorly qualified or briefed, or in which the experts were probably recommended by the developer, thus giving rise to a potential conflict of interest.

### 3.6. The Environment Agency (EA) and the Scottish Environmental Protection Agency (SEPA)

The EA (the Scottish equivalent being SEPA) has expertise in the shallow (<500 m deep) geology, because its remit extends to protection of groundwater resources. But it is also aware, in a general sense, of the environmental risks of unconventional hydrocarbon exploitation (Appendix A). It is a statutory consultee of the MPA. Naturally the EA leans heavily on the BGS for geological information; indeed, many published EA reports are jointly authored by the BGS.

It is of concern that part of the EA’s income depends upon the grant of permits, thereby entailing a conflict of interest. A precautionary approach may be subsumed under the pressure to generate income.

The case histories presented below show that, whereas there is no doubt that the officials of the two organisations are acting in good faith in the interests of the public and of the environment in general, permits have at times been granted based on a limited or out-of-date understanding of the pertinent geology.

### 3.7. The Health and Safety Executive

The Health and Safety Executive (HSE; there is a separate HSE for Northern Ireland) is a statutory consultee of the MPA. In the context of geology, its remit is to advise on well construction, and to monitor the wellsite during and after drilling and completion. While its geological role is limited relative to the other agencies, there are several examples of inadequate oversight and/or record-keeping by the HSE which have proved to be geologically pertinent.

### 3.8. The British Geological Survey

The British Geological Survey (BGS) is a component body of the Natural Environment Research Council. It is not a statutory consultee, and has no formal role in the grant of permits and licences. Nevertheless, as the national repository of geological information, it has an important background role. It collaborates with the EA and SEPA, and publishes joint reports with those agencies.

The BGS claims under its current logo to be “Expert, Impartial, Innovative”. Unfortunately, an institution such as the BGS, half-funded by NERC (and therefore indirectly by central government), can never be fully impartial. Its working earth scientists, of whom I was one for fourteen years, may have complete scientific integrity, but what they are permitted to work on and publish is controlled by managers higher up the chain of command. The managers answer to government departments, or ‘customers’, to use the jargon introduced by the Rothschild report of 1971 into the funding of UK civil science. So, the supposedly impartial science emanating from the BGS may on occasion be incomplete, or even misleading. Several examples are provided in the Appendix A: Case histories of relationship between BGS and government.

The BGS was commissioned by DECC to write two important papers, one for each of the areas in England affected by unconventional hydrocarbon activity, the Bowland Shale of the north of England [37] and the Jurassic shales of the Weald [38]. A senior BGS scientist also wrote a non-technical book on the science of fracking [39], ostensibly independently of the BGS, but presumably with the sanction and support of the BGS and government.

### 3.9. Ad Hoc Expert Groups, Interested Bodies, Consultations

A joint review of fracking for shale gas was undertaken by two UK academic societies in 2012, the Royal Society and Royal Academy of Engineering [40] (‘RR2012’). It concentrated on the risk of induced seismicity, but failed to address the problem of through-penetrating faults in the UK shale basins. The problem of pre-existing faults was barely discussed, even though it had been introduced as a subject for concern by a submission to the expert committee from the Geological Society of London. Instead, the report accepted uncritically the conclusions of a Halliburton study [41] (a major US oil industry service provider) as did a report [42] commissioned the same year by DECC. The presentation by the BGS to one of the RR2012 evidence sessions [31] also failed to mention faults, relying instead on oversimplified geological cross-sections more applicable to the USA than to the UK, and quoting in positive terms from the Halliburton study.

RR2012 made ten recommendations, which were all accepted by government; but only one of them, the traffic light system for mitigating induced seismicity, was ever implemented—and that was done in an oversimplified way.

An Independent Expert Scientific Panel was set up by the Scottish Government in September 2013, and reported in July 2014 [43]. Its remit excluded the making of recommendations to the Scottish government. Its cautious findings were summarised in fourteen main points. It concluded that safe unconventional exploitation could be developed “subject to robust regulation being in place”, and:

“The regulatory framework is largely in place to control the potential environmental impacts of the production of unconventional oil and gas in Scotland, although there may be gaps to address”.

Consultations are an integral part of the planning and regulatory process, and provide the main vehicle for local interest groups, objectors, and so on to express their views. The important role played by such groups has been recounted in two books by Alan Tootill [44,45], and in academic studies [46]. However, the examples below will show that when geology is introduced into such consultation responses, whether to an MPA or to the EA/SEPA, it has invariably been ignored or given a low weight.

The OGA hosted a one-day seismicity workshop on 3 October 2018 to investigate the Newdigate earthquake swarm. This is described in Section 4.13.2 below.

A joint Shale Environment Regulator Group (SERG) was set up in October 2018, to coordinate the regulatory responsibilities of the OGA, the EA and the HSE [47]. There is no evidence to date that SERG has ameliorated the regulatory system.

## 4. Case Histories

### 4.1. The Weald: Introduction

The Weald Basin is located in Southern England (Figure 2). It comprises the central and eastern part of a larger Wessex–Weald Basin, some 200 km in length, aligned east–west over Southern England from Wiltshire in the west to Kent in the east (Figure 3a). The principal target of unconventional exploration onshore in Southern England is the Kimmeridge Clay Formation of Jurassic age, the mature area of which is shown by the hatching in Figure 3b (*mature* means developed by pressure and temperature over geological time such that hydrocarbons have been created). This is a well-known source rock in the Weald, in Eastern England, in the Hebrides, and in the North Sea. The wider Weald–Wessex Basin was thoroughly explored up to the 1980s.

In response to the renewed interest in unconventionals the BGS was commissioned by DECC to produce a report [38] on the Jurassic of the Weald Basin. The main characteristic of the new phase of exploration, which started with DECC awarding PEDLs in the 13th onshore round of 2008 [48], is the fact that most of the companies in the Weald have tried to hide behind a façade of conventional exploration, while pursuing an unconventional programme. The oil majors are no longer interested in the Weald; new exploration is now dominated entirely by ‘penny share’ start-ups.

### 4.2. The Weald: Cuadrilla, Balcombe, 2008–2014

The area of thick mature Kimmeridge Clay Formation in the Weald Basin is shown in Figure 3b by the cross-hatching. Cuadrilla Resources Limited (CRL) obtained the PEDL 244 licence in September 2008. In its planning application of 25 January 2010 to West Sussex County Council (WSCC) it declared its intention to frack at Balcombe (Figure 3c; see also Appendix A: Cuadrilla at Balcombe). It described the proposed fracking process, while avoiding mention of the words ‘hydraulic’ or ‘fracturing’. The application was granted [49] on 23 April 2010, with conditions. 

A letter from CRL to DECC dated 10 June 2011 stated, more explicitly about the unconventional nature of the drilling:

“In order for Bolney [the subsidiary of CRL] to be successful in its Weald Basin Kimmeridgian Oil Shale project (KOSP), Bolney will need to rely, to a significant degree, on being able to undertake *hydraulic fracture*
*stimulation(s)* of this unconventional reservoir” (emphasis added).

By May 2013 Cuadrilla had changed its position on fracking, assuring Balcombe Parish Council [50] (see Appendix A: Meeting Balcombe PC and Cuadrilla 2013) that fracking would not be part of the testing procedure of the proposed vertical and horizontal wells; instead “They will ‘stimulate’ the reservoir rock using low pressure hydrochloric acid in a concentration between 7.5% and 15% that they say is classified as non-hazardous”—in other words, acidisation [12]. However, Cuadrilla did leave open the possibility of fracking in the future:

“If there was insufficient natural flow consideration would be given to whether or not the reservoir rock could be safely fracked and if so the necessary environmental impact assessment and planning permissions might be sought to Frack and flow test the well (this would be a separate and future operation if approvals were sought and granted)”.

The EA issued a permit on 24 July 2013 for Cuadrilla to drill horizontally some 700 m along the mid Kimmeridgian ‘I’ micrite. But the WSCC planning permission expired in September 2013, after the vertical and horizontal wells had been drilled but before testing had been carried out, so Cuadrilla sought a six-month extension period to enable it to complete testing.

A team from the University of Bristol deployed four seismometers around the Balcombe site to measure microseismicity [51] between July and September 2013, spanning the drilling period. The source of funding for this small project is not disclosed in the Bristol paper; however, the paper is incorporated as Appendix O to the Balcombe-2 and -2z well data, and referred to in the joint end-of-well report as follows:

“As part of Cuadrilla’s on-going seismic monitoring programme, 4 seismometers were deployed around Balcombe-2/2z, all of which were provided by Bristol University”.

The vertical well Balcombe-2 was spudded on 2 August 2013. The horizontal well, Balcombe-2z, was drilled as a sidetrack, and the well was suspended just before its planning permission expired. New approvals were sought for testing. Angus Energy acquired the licence from Cuadrilla in January 2018. Its proposed three-year-long test proposal was recommended for rejection by the MPA in April 2020. The application was withdrawn just before it could be determined, and has been replaced by a new application, validated on 26 August 2020, for a 55-week extended well test [52].

### 4.3. The Weald: Celtique Energie, Fernhurst and Wisborough Green 2014

In 2013 Celtique Energie Limited proposed two well locations, Wisborough Green (WG) and Fernhurst (F), shown by the red crosses in Figure 3b. The company proposed to drill horizontally along one of the two Kimmeridge micrites, just as Cuadrilla had done at Balcombe. The Wisborough Green planning application was submitted to WSCC [53] (the local MPA), whereas the Fernhurst application can be found on the South Downs National Park Authority (SDNPA) planning website [54]. SDNPA is the MPA for the South Downs. The applications did not include fracking of the horizontal wells, but the company stated that fracking might prove to be necessary, and that it would in that case submit additional applications. I submitted formal objections both to the Fernhurst [55] and to the Wisborough Green [56] applications.

Celtique submitted the same 8 km long sample of 2D seismic data in support of both applications, even though the sites are 16 km apart. The location of the sample line is shown by the wavy green line in Figure 3b. Its significance is discussed below. Both applications were refused. Fernhurst was turned down by the SDNPA on 11 September 2014, principally on the ground that Celtique had not justified why it needed to drill within the National Park. Therefore, it may be concluded that geology did implicitly play a part in its decision, in that the target shale formation is found throughout the PEDL, so the MPA could justifiably say that the target should be sought outside of the National Park.

### 4.4. Scotland Coal Bed Methane: Canonbie

The concealed Canonbie coalfield [57] was mentioned in a DECC report [58] aimed at promoting unconventional exploration of CBM. Figure 2 shows the location of the area, which straddles the Scotland–England border.

PEDL 159 was awarded to Greenpark Energy Limited in the 12th onshore licensing round of 2004 [59], with a commitment to drill one vertical and three deviated horizontal wells; seven wells were then drilled by Greenpark. Dart Energy’s subsidiary GP Energy Limited acquired the licence in 2012, and the licence was relinquished on 31 August 2015. Some of the exploratory sites, marked by blue diamonds in Figure 4, lie over the Coal Measures of the Canonbie coalfield at outcrop, whereas thirteen lie on the primary aquifer of the Sherwood Sandstone Group or the underlying Permian.

In early 2015 the *Sunday Herald* [60] revealed that Dart Energy had, firstly, received permits for the drilling of 18 separate holes from the Scottish Environmental Protection Agency (SEPA), and secondly, that the permits had been renewed in 2011 and 2013, despite the discovery that four of the boreholes had not been constructed in a manner to prevent pollution of the shallow groundwater resource. An internal SEPA report [61] stated: 

“four wells were constructed with casing that was not cemented between 100 m and 400 m below ground level. This potentially allowed saline waters from the Coal Measures at the bottom of this uncemented zone to travel up to and contaminate the Permian Sandstone aquifer at the top of this zone”.

The report recommended escalation of the control of deep borehole drilling authorisation “from GBR to a complex licence”. The General Binding Rules (GBR):

“represent a set of mandatory rules which cover specific low risk activities. Activities complying with the rules do not require an application to be made to SEPA, as compliance with a GBR is considered to be compliance with an authorisation. Since the operator is not required to apply to SEPA, there are no associated charges”.[62]

Independently of, but contemporaneous with, PEDL 159, New Age Exploration Limited was granted an exploration licence [63] at Canonbie by the Coal Authority in June 2012. Its so-called Lochinvar acreage overlaps with PEDL159. There is no public record of any interaction between the two regulators (DECC and the Coal Authority), even though both were licensing the same geological target, the coal seams of the hidden Canonbie coalfield (see Appendix A: Canonbie).

### 4.5. Scotland Coal Bed Methane: Airth 2014

#### 4.5.1. History

Tricentrol Exploration Limited held licences in the 1980s to explore for oil. Interest in coal bed methane started with Hillfarm Coal Company in 1993. Coalbed Methane Limited proposed a pilot project in the Airth area in 1995 (located in Figure 2). Coalbed Methane Ltd. took over the licence (EXL237) by 1996 and drilled the Airth-2, -3 and -4 wells. Composite Energy took over Coalbed Methane’s interests by 2004, obtained planning permission in 2005 (the licence now being PEDL 133), drilling Airth-5-11 inclusive by 2007 as horizontal completions. Dart Energy acquired Composite in 2011 and submitted a planning application for a Coal Bed Methane (CBM) development near Airth in August 2012. The proposal was for well site establishment at 14 locations and development of inter-site connection services, site access tracks, a gas delivery and water treatment facility, ancillary facilities and infrastructure and an associated water outfall. Falkirk and Stirling District Councils failed to reach a decision on the application, and in July 2013 Dart appealed to the Scottish government for a decision. A public inquiry was set up in August 2013, in response to 2500 objections to Dart’s proposals, and opened in March 2014.

The period of the inquiry overlapped with that of the Scottish Independent Expert Scientific Panel [43]. The determination of the Airth inquiry was deferred by the moratorium on unconventional oil and gas (UOG) introduced in January 2015. UOG was deemed to include CBM. A public consultation was launched in January 2017. A policy of no support for UOG was announced in the Scottish parliament on 3 October 2017, and finalised [64] on 27 February 2020. INEOS, now the licensee of PEDL133, tried to re-open the inquiry earlier in February 2020, but a month later finally withdrew the planning application [65]. The inquiry report was called in and to date remains unpublished.

#### 4.5.2. Dart Energy at Airth

In contrast to its strategy at Canonbie, where separate apparently unconnected planning applications were put in, Dart presented a comprehensive programme of work at Airth as one application.

My submission to the Airth inquiry of 2014 outlined a number of erroneous or misleading interpretations of the geology presented by Dart in its planning application. These are summarised in the Appendix A: Airth 2014, but there is no merit in presenting them herein because it is not yet known what weight, if any, was given to these issues by the Inquiry Reporter.

### 4.6. Lancashire 2009–2013

An application to drill at Hale Hall near Kirkham in the Fylde was prepared by Cuadrilla in late 2009, but the site was never drilled. Figure 2 shows the location of the Fylde area of NW Lancashire.

Preese Hall-1, a vertical well, was spudded on 16 August 2010, and drilling was completed on 12 December 2010. Fracking of the shale in Preese Hall-1 in 2011 triggered earthquakes, resulting in a moratorium. This induced seismicity is discussed below.

Grange Hill was drilled between 15 January and 13 May 2011, according to the dates given in a subsequent planning application by Cuadrilla, which stated [66]:

“The drilling operations overran slightly due to the hard formations encountered … Well testing operations at Grange Hill were not carried out due a moratorium on fracturing imposed by the Department of the Environment and Climate Change (DECC). Operations ceased at Grange Hill in May 2011 and the site has been retained, with a suspended well, awaiting further announcements from DECC”.

This statement contradicts the record in the composite well log, which agrees with the start date of 15 January, but states that drilling was only completed on 19 July 2011. The sidetrack 1z was drilled to bypass a drilling assembly stuck in the original borehole which could not be recovered. The wells were not fracked and there was no flow testing of gas.

The Anna’s Road-1 well was spudded on 6 October 2012, and drilling was completed on 12 November 2012. It bottomed in the Sherwood Sandstone Group at 620 m below sea level. Cuadrilla reported [67] that a packer had become stuck at the bottom of the well, and that the well would be re-spudded 3 m away from the original because a packer cannot be drilled through. But this re-spud never took place. A year later the site was abandoned.

All three wells were planned and drilled on geological information interpreted from the 2D seismic database, before the results of the 3D survey undertaken by Cuadrilla in mid-2012 became available. It can be seen in hindsight that if the 3D seismic survey had been undertaken and interpreted first, the geological problems with these wells might have been avoided. But regulation does not stipulate that such a survey must pre-date any drilling.

### 4.7. The Weald: 2014 to Date

#### 4.7.1. Kimmeridge Clay Formation

The BGS Weald shale study [38] of 2014 states, on the Kimmeridge Clay Formation (KCF):

“Argillaceous rocks are dominant, with some being organic-rich, although there is a paucity of ‘hot shales’ with high gamma-log peaks in the Weald area…The thickest well penetration is 1864 ft (568 m) in Balcombe 1 …. Several coccolith micrite beds are present within the Kimmeridge Clay, notably in the eastern Weald…. where they are known as the mid-Kimmeridge micrites. The lower ‘J-Micrite’ reaches a maximum thickness of c.125 ft (38 m), whereas the upper ‘I-Micrite’ is up to 150 ft (45 m) thick. A thinner ‘K-micrite’ has a more restricted distribution. The micrite beds thicken towards the basin centre and pinch out towards the basin margins. These low porosity and low permeability micrites may be targets in a hybrid Bakken-type shale play, with shale units above and below. The oil in the micrite in the Balcombe 1 well has been compared to the hybrid Bakken play”.

The micrite beds are calcareous mudstones or ‘dirty’ (impure) limestones, known informally in the hydrocarbon industry as micrites. The difference between the different numbers of micrite layers recognised within the KCF is partly one of interpretation. The Balcombe-1 log illustrates the problem (see Appendix A: Micrites in the Kimmeridge Clay Formation). The log shows that the micrites cannot be classed as ‘limestones’, the term misleadingly employed by the industry.

#### 4.7.2. Angus Energy, Brockham

BP drilled Brockham-1 in 1987 (Br in Figure 3b above), testing oil from the Portland Sandstone in a tilted fault block trap. This discovery was named the Brockham oilfield. Several more wells were drilled between 1998 and 2007 under a variety of operators succeeding BP. Angus Energy drilled Brockham-X4z in early 2017, sidetracked from Brockham-X4, which had been drilled a decade earlier.

The regulatory issues arising since Angus acquired the licence (now production licence PL235) in 2012 include:Appraisal or production?Conventional or unconventional?Was Brockham-X4z drilled without planning permission?

These issues are discussed below.

#### 4.7.3. UKOG Broadford Bridge

Celtique Energie identified the ‘Willow Prospect’ in 2012 in PEDL 234. This is a conventional hydrocarbon trap, with the reservoir being prognosed as Sherwood Sandstone (Triassic age) at 7000–8000 ft (2.1–2.4 km) depth. It lies to the north of, and is bounded by, the Broadford Bridge Fault (Figure 5). The proposed site was one of seven possibilities examined in the Alternative Sites Assessment. Because the trap is finite in extent there is a limited area within which surface sites for drilling may be searched for.

UKOG took over PEDL234 from Celtique in June 2016. It intended to explore for hydrocarbons in a completely different manner. Firstly, in contrast to Celtique’s well-defined conventional target, the Applicant’s target, the Kimmeridge Clay Formation (KCF), is found below the whole of the licence area. The Initial Drilling Application [68] to the OGA confirms that the target was anticipated to be a “resource or continuous oil deposit, without a structural closure”. Therefore, there was no geological requirement or justification for using the existing well pad at Wood Barn Farm. It also follows that the Alternative Sites Assessment carried out by Celtique, which was a material part of the planning approval, was superfluous, since the entire KCF over the whole licence block had become the target.

The OGA issued a drilling permit to UKOG on 25 May 2017. This included permission to sidetrack. I provided detailed evidence to the EA consultation that UKOG was planning to breach the conditions of the permit issued by the MPA. Nevertheless, the EA issued a permit on 6 July 2017.

Broadford Bridge-1 was spudded on 29 May 2017, before the EA permit was issued, and completed on 29 July, the same day that the sidetrack Broadford Bridge-1z was begun. The sidetrack was completed on 1 September 2017, but there was a problem with the cement bonding of the casing [69] and reported formation damage [70]. Work stopped in March 2018 and the well was suspended. In July 2020 a third time extension was granted by the MPA for two more years, during which UKOG said it would not carry out any new work, but would compare the Broadford Bridge results with its other sites in Surrey [71].

#### 4.7.4. UKOG Horse Hill

PEDL137 was awarded to Magellan Petroleum in the 12th onshore licensing round in 2004. A planning application to drill Horse Hill-1 (labelled HH in Figure 3b) was submitted Surrey County Council (SCC; Mineral and Waste Application RE10/2089) in 2010. Planning permission to drill was granted by on 16 January 2012 following a favourable Officer Report dated 9 November 2011. The Officer Report summarised the geology behind the proposal as follows:
“The Horse Hill Prospect has been identified through seismic survey and interpretation of the drilling from Collendean Farm 1. Drilling would target the Portland Sandstone, Corallian Beds in the Jurassic Formation and the deeper Triassic Formation, which has a predicted top of formation at 7,300 ft True Vertical Depth Sub Sea (TVDSS). It is proposed to drill to a total depth of 7,483 TVDSS. The applicant has stated that the Portland Sandstone and Corallian Beds have been shown elsewhere as productive in relation to oil and that gas flows have been recorded from the upper Triassic Formation”.
and:
“The current application involves the drilling of a well to potential target areas: the Portland and Corallian sandstones which the applicant expects to be oil bearing at this location and the Triassic which the applicant expects to be gas bearing”.
and
“The applicant does propose a deviated well to access three target areas: The Portland, the Corallian and the Triassic at approximately 499 m, 1143 m and 2143 m depth respectively. The initial geological target, the Portland Target, is relatively shallow and therefore the well deviation would not begin until close to the base of the Portland Sandstone at approximately 671 m. From that point the borehole would build angle, but to minimise the chance of difficulties with the wellbore, the build rate is programmed not to exceed 3.0° per 30.5 m up to a maximum angle of 22.6°”.

None of the planning documents mention unconventional exploration, nor is the Kimmeridge Clay Formation mentioned. However, there were no particular conditions attached to the planning permit in regard to drilling to the targets specified above.

Horse Hill-1 was spudded on 24 September 2014 by Horse Hill Developments Limited (HHDL), a special purpose company, which took a 65% interest in the prospect, leaving Magellan with 35%. It proved a minor oil discovery in the Portland Sandstone, but received more publicity in February 2015 when UKOG (a partner in HHDL) announced that a 400 m vertical thickness of the Kimmeridge Clay Formation had a very high organic content.

Flow testing of the well began in February 2016. A planning application for further appraisal and for drilling of two more wells was submitted to SCC in October 2016. It was approved on 18 October 2017. In December 2018 plans were submitted for four more wells and twenty years of production at Horse Hill. Planning permission was granted by SCC in September 2019.

#### 4.7.5. Europa Leith Hill

Europa Oil & Gas (Holdings) plc (‘Europa’) was awarded the PEDL143 licence in the 12th onshore licensing round in 2004. The targets in the ‘Holmwood’ prospect were the Portland Sandstone (two horizons) and the Corallian Sandstone. It later added Kimmeridgian micrites to the list of targets. The Applicant asserted that the drilling operation was conventional in nature. The Holmwood prospect dates back to its identification by BP in the 1980s, but has never been drilled (see development timeline and geological details at Appendix A: Europa at Leith Hill).

Europa stated in its application for the PEDL (Europa Oil and Gas Limited 2004):

“Assuming success with the planning process for the well, and more crucially an indication that planning permission would be forthcoming for any future development, Europa plan to acquire two new vibroseis 2D seismic lines and drill the Holmwood Prospect, testing both the Portland and Corallian targets. A commitment would be made by the end of the 3rd year of the licence to complete a well to test both Protland (sic) and Corallian levels by the end of the licence term”.

The final terms and conditions of the PEDL award are not available; however, it would be surprising if DECC had waived the offer to acquire the additional seismic data, a work commitment which is at the bare minimum of what is generally considered acceptable for obtaining a PEDL. Naturally, the new seismic data would (and should) have been acquired before drilling site selection. However, no additional seismic data have ever been acquired.

The Holmwood prospect lies south of Leith Hill in the Mole Valley, but the drill pad (labelled LH in Figure 3b) lay on the dip slope of Leith Hill at Bury Wood, leased from the Forestry Commission. This lies some 1.5 km NNE of the centre of the prospect. However, the lease expired in September 2018 and was not be renewed, so Europa closed the site, and in March 2019 transferred the operatorship of PEDL143 to UKOG.

### 4.8. Lancashire 2013 to Date

#### 4.8.1. Roseacre Wood

Cuadrilla unveiled plans to develop two new sites in early 2014 at Roseacre Wood and Preston New Road (Figure 6).

In October 2016, the Secretary of State said that he was minded to allow the appeal relating to the Roseacre Wood site, but that the public inquiry should be reopened to allow further evidence to be submitted by the appellant in relation to highway impacts. The reopened public inquiry took place in April 2018. On 12 February 2019, the Secretary of State refused the appeal, concluding that safe and suitable access to the site would not be achieved, and that significant impacts from the development on highway safety would not be mitigated to an acceptable degree. There would be an unacceptable impact on highway safety. Cuadrilla surrendered its permit in August 2019. Although the site was never drilled, the regulatory decisions of the EA are worth noting, and are discussed below.

#### 4.8.2. Barton Moss

In June 2010 Salford City Council granted Nexen Exploration UK Limited planning permission for “Drilling of two exploratory boreholes for coal bed methane and production” in PEDL193 at Barton Moss near Eccles, Manchester. Figure 2 shows the site location. IGas took over the licence from Nexen and drilled Irlam-1 and Irlam-1z, in early 2014.

#### 4.8.3. Preston New Road (PNR)

On 6 October 2016, the Secretary of State for Communities and Local Government allowed Cuadrilla’s appeal against the County Council’s decision to refuse permission for fracking at the Preston New Road site and granted planning permission, subject to conditions. Construction of the wellsite started in January 2017, and two wells were drilled between August 2017 and July 2018. Hydraulic fracking consent to fracture these wells was granted by the Secretary of State in August/September 2018.

A permit [72] had been issued by the EA in January 2015 for (inter alia) monitoring of induced seismicity, comprising a ten-station surface monitoring array, and an 80-station array of buried near-surface seismometers. However, in June 2017 Cuadrilla requested a variation of the permit, replacing the 80-station set-up by a downhole array in a nearby borehole. Despite objections, this variation was accepted by the EA. The implications of this approval are discussed in Section 5.12.6 below.

PNR-1 was spudded on 16 September 2017 and completed on 11 January 2018, the same day that the sidetrack PNR-1z was started. PNR-2 was spudded on 17 August 2017. A significant discovery revealed by PNR-1 was that the predicted layer of 300 m of Millstone Grit was entirely absent. The consequences of this absence are discussed in Section 5.12.8 below.

Fracking of PNR-1z took place between October and December 2018. Fracking of PNR-2 started on 15 August 2019. The ensuing seismicity caused by each phase of fracking is discussed in Section 4.13.3 below.

### 4.9. East Midlands: IGas, Springs Road, Misson

IGas Energy took over Dart Energy in May 2014. A timeline of IGas exploration [73] at Springs Road, Misson, Nottinghamshire (located in Figure 2), is as follows:15 October 2015: application submitted to drill one vertical and one deviated well.15 January 2016: application to the Environment Agency for a mining waste permit.16 June 2016: EA permit granted.15 November 2016: Nottinghamshire County Council’s planning committee voted to approve the application.2 June 2017: legal agreement published.22 January 2019: first (vertical) well Springs Road-1 spudded.23 March 2019: Springs Road-1 completed.

The approval was conditional on a legal agreement including requiring IGas to pay a bond that would cover the cost of site restoration, in view of local concerns about the company’s financial viability [74]. Work could not start until the first deposit of GBP 120,000 of the bond had been made. A second payment of GBP 290,000 was due before vertical drilling could start, and horizontal drilling could not begin until after the final payment of GBP 240,000.

To date the horizontal (deviated) well has not been drilled.

### 4.10. Cheshire: Ellesmere Port

Nexen Corporation applied for planning permission in October 2009 to drill an exploration well for coal bed methane within the Coal Measures at Ellesmere Port in PEDL184, with a requirement to drill to a minimum depth of 900 m. The location is shown in Figure 2. Ellesmere Port-1 was drilled between November and December 2014 by IGas, but to a depth of some 1950 m (measured depth).

On 25 January 2018 the MPA refused planning permission for tests on Ellesmere Port-1, drilled in 2014. IGas appealed against the decision. A public inquiry was held in early 2019. It was called in by the Secretary of State, but the determination has not yet been made.

### 4.11. DECC/OGA Selective Licence Offers in SE England

DECC’s 14th onshore round of licensing of 2015 was designed to open up for exploration all the areas which have even the slightest potential for shale gas or oil. Basically, this means the whole of the UK, except for the mountainous and upland areas where there is crystalline rock at the surface, and no prospect whatsoever for shale fracking. So the Highlands and Southern Uplands of Scotland were omitted, together with the Lake District, most of Wales, and Cornwall. Figure 7a shows the geology of southern England in a variety of colours, with the 14th round offer areas and existing wells on top. The parliamentary constituency of Witney is shown in solid blue. David Cameron, UK prime minister 2010-2016, is the member of parliament for Witney.

The surface geology is shown in colour in Figure 7a, and although this picture is not necessarily a reliable guide to the geology at depths of more than a kilometre or so, we know that the upper layers of rocks in the south-east are of Mesozoic and Tertiary age, and that these contain several important shale and clay layers, possibly suitable for fracking. The Witney constituency is no exception. Possible reasons for the omission are discussed in Section 5.14 below.

### 4.12. DECC/OGA Lax Licence Offers

In July 2016 Drill or Drop [75] summarised the OGA grant of licence extensions by PEDL number as follows:Initial term extensions for one year: 224, 241, 253.Initial term extensions for two years: 143, 162, 164, 189, 204, 214-217, 233-235, 254.

Subsequent extensions to licensing terms have since been requested as follows (information from Drill or Drop [76]):April 2017: UKOG granted extensions to its retention areas at Horse Hill.August 2017: a third extension was awarded to the term of Dart Energy’s PEDL189. A High Court appeal against the extension was rejected.July 2018: Europa sought three more years at Leith Hill, a second PEDL extension, plus other extensions.July 2018: INEOS sought and obtained a second licence extension in PEDL162.January 2020: UKOG asked for two additional years at Broadford Bridge, the third extension requested. The company blamed slow progress at Horse Hill.July 2020: Egdon asked twice for an extension to its licence dating from 2014. The only works completed to date have been at the site entrance and a layby in the lane.

In the PEDL189 case above, the judge accepted the Secretary of State’s argument that the PEDL was a contract, and that the terms could be varied by agreement between the parties, like any other contract. He concluded that a PEDL licence was not governed entirely by the statutory code (see also Section 3.4 above).

INEOS provides an example of an operator being unable to fulfil its licence obligations. This is discussed in Section 5.15 below.

Some PEDLs have been awarded when the work commitment is derisory, such as one conditional well being offered, or a minimal amount of 2D seismic data being acquired. There is no sanction for non-fulfilment of licence obligations.

### 4.13. Earthquakes and Induced Seismicity

#### 4.13.1. Preese Hall-1, Lancashire

The Preese Hall-1 vertical well was drilled into the Bowland-Hodder unit in 2011 in the Fylde, Lancashire (Figure 2). Fracking was undertaken. It transpired that the basin is near-critically stressed, and the fracking triggered a series of small earthquakes. The shale is gas-prone and over-pressured. Three separate studies of the earthquake triggering problem were then undertaken [42,77,78]. The last authors mapped the locations of the seismic events induced by the fracking, and concluded that they all occurred on a single pre-existing fault favourably aligned to the regional horizontal principal compressive stress component.

#### 4.13.2. The Newdigate Earthquake Swarm

The OGA hosted a one-day seismicity workshop on 3 October 2018 to investigate the Newdigate earthquake swarm in Surrey. The area is cut by normal faults with displacements of the order of tens to hundreds of metres. There has been no active tectonism in the region since the mid-Miocene uplift and tectonism related to Alpine compressional events. The swarm had abruptly started on 1 April 2018, as felt events causing visible damage, after centuries of aseismicity in the area. The invited attendees included four BGS scientists, five academics, eight government employees, three industry representatives, plus a representative from the MPA, Surrey County Council. The OGA report of the workshop concluded [79] that:

“based on the evidence presented, there was no causal link between the seismic events and oil and gas activity although one participant was less certain and felt that this could only be concluded on ‘the balance of probabilities’ and would have liked to see more data on nearby oil and gas surface activity over the past two years …”.

The swarm is discussed in Section 5.9 below.

#### 4.13.3. Preston New Road, Lancashire

Fracking of PNR-1z started in October 2018, with the downhole seismic monitoring array placed just above the heel of the adjacent PNR-2. Sixty-nine seismic events with local magnitude M_L_ > 0.0 were triggered, the largest being M_L_ = 1.5 on 11 December 2018. Four events were of magnitude exceeding the 0.5 threshold in the traffic light system, requiring Cuadrilla to halt operations temporarily.

Fracking of PNR-2 started on 15 August 2019, immediately triggering seismicity. Fracking was suspended by the OGA on 26 August after the largest event to date, M_L_ = 2.9, occurred, to allow time for a review of the hydraulic fracture plan.

On 2 November 2019 the government ordered a moratorium on fracking in England [80] because of the risk of earth tremors. Governments in Scotland, Wales and Northern Ireland had already adopted measures that amount to moratoria on fracking.

## 5. Discussion

### 5.1. Exploration or Appraisal?

The shale wells are frequently described as exploratory. But they do not conform to the modern definition of an exploratory well. For example, the US Securities and Exchange Commission, in seeking to modernise definitions to increase business transparency and reporting [81], defines an exploratory well as follows:

“An exploratory well is a well drilled to find a new field or to find a new reservoir in a field previously found to be productive of oil or gas in another reservoir”.

The Norwegian Petroleum Directorate clarifies and expands on the definition [82]:

*“Exploration well:* a well drilled in order to establish the existence of a possible petroleum deposit or to acquire information in order to delimit an established deposit. Exploration well is a generic term for wildcat and appraisal wells.*Wildcat well:* exploration well drilled to establish (prove) whether petroleum exists in a potential petroleum deposit.*Appraisal well:* exploration well drilled to establish the extent and size of a petroleum deposit that has already been discovered by a wildcat well”.

Note that the Norwegian definitions define Wildcat and Appraisal wells as sub-categories of Exploration wells. The precise categorisation of well type has not mattered too much up till now, because in conventional hydrocarbon exploration it is usually obvious to which category a well belongs. In true wildcat drilling the exact location of the well can be all-important. Target locations are specified to the nearest metre, both horizontally and in depth. There are many case histories of wells failing (coming in dry) because the oil-water contact was missed, or the well went down on the wrong side of a fault, perhaps by a matter of a few metres.

However, unconventional oil and gas exploitation is more of an industrial extraction process than a genuine ‘exploration’ process to find an economic resource that may or may not be present, and this fact is changing the nature of the exploration industry. There is negligible oil or gas flow from a shale, which is a ‘tight’ (i.e., extremely low permeability) rock; that is, it only yields up its oil or gas very slowly, over geological aeons. There is no shale from which, once drilled, the oil or gas flows freely under its own pressure. If there were, then shales would have been drilled routinely to extract conventional oil and gas, and the entire hydrocarbon exploration industry, devoted over the last century to finding progressively more subtle and obscure reservoirs, would never have developed in the way that it has, because shales comprise about half of the volume of rock in a given sedimentary basin. One would simply drill down to the nearest shale layer and extract the hydrocarbons. Therefore, all shale drilling should be classified as Appraisal, or, further down the line, as Development and/or Production.

### 5.2. Canonbie

At Canonbie we see two independent licensing authorities, DECC (succeeded by the OGA) and the Coal Authority, acting in apparent ignorance of each other’s regulation. It is evident that the same coal seams cannot both be drilled for methane gas extraction while being mined to extract the coal.

The tactic used by Dart Energy at Canonbie is an example of ‘salami-slicing’; that is, it treated each borehole planning application as a separate small development. SEPA was apparently unable or unwilling to see that the projects were part of a single larger scheme encompassing more than 20 km^2^. 

The drilling of four boreholes without adequate protection of the groundwater resource illustrates the lack of responsibility and foreknowledge on the part of SEPA; however, the authority did realise some three years after the event that the boreholes were non-compliant [83] (see Appendix A: Minutes of SEPA Enzygo Greenpark Meeting 2011). The SEPA guide to General Binding Rules states, in effect, that allegedly low risk environmental activities do not need specific authorisation—an extreme example of a laissez-faire approach to environmental protection. Rob Edwards’s report [60] of 2015 also illustrates the excuses and buck-passing from one agency to another:

“Sepa pointed out that gas boreholes can’t be used unless it grants them a licence. After discussions with the Health and Safety Executive and the Department of Energy and Climate Change in London, it had strengthened the regulatory requirements from April 2013.

‘*The local authority is responsible for granting planning permission for surface works associated with borehole construction, but is not involved with the design or sign off of the borehole itself*’,said a Sepa spokeswoman

Dumfries and Galloway Council said it couldn’t comment while the matter was under investigation by the ombudsman. ‘*The alleged non-compliance with the Sepa licence is a matter for Sepa and it would have no impact on the council’s interests as planning authority*’”,added a council spokesman

The SEPA excuse given above is self-serving and irrelevant, because the fact is that several boreholes existed for several years as potential conduits for pollution of potable groundwater, whether or not gas extraction ever took place. It is stated that discussions eventually took place between SEPA, the HSE and DECC to resolve the problem. However, one of these authorities should have had clear responsibility for borehole design and enforcement before any drilling was permitted.

One of the proposed sites, at Becklees Farm, lies across the border in England (Figure 4), and is therefore under the jurisdiction of the EA, not SEPA. Greenpark Energy [84] stated:

“The site is situated over an area of Triassic and Permian Sandstones which act as a major aquifer. However, because there is a layer of overlying drift, the Environment Agency have designated the local groundwater as having a low vulnerability to potential pollution because of the protective properties of the overlying drift”.

The site is located where the major aquifer is overlain by a mere 5 m of ‘overlying drift’ (post-glacial till) [85]. The only conceivable explanation for the EA approval is that it was narrowly considering the risk of pollution from spills at the surface possibly penetrating down to the aquifer, and did not consider upward migration of pollutants from depth into the aquifer, via the drill bore and/or faults. This is a severe lapse in understanding and regulation.

The EA appears to have insufficient in-house geological expertise to respond to planning applications in this area. However, instead of strengthening the expertise, government implemented a 15% staff job cut. The EA asserted, furthermore, that from February 2014 permits would be issued within 1–2 weeks. It is difficult to see how this haste could be reconciled with “taking into account the views of local communities, environmental organisations and other stakeholders” [86]. 

Canonbie demonstrates that the regulatory regime had become, in effect, one of self-regulation. 

### 5.3. The Weald: Cuadrilla, Balcombe

Figure 3b shows the Balcombe wells (Bal) in the area of mature Kimmeridge Clay (hatched area, as defined by the BGS [38]) within the Wessex–Weald Basin (Figure 3a). Existing wells are shown by red dots. Figure 3c shows the Balcombe locality. Here the mapped surface-outcropping normal faults are shown by red lines with teeth on the downthrown side. The green line is the CDP location of 2D seismic line TWLD-90-27, dating from 1990, which was used in Cuadrilla’s planning application without having been reprocessed. There is another, even older, seismic line coincident with this one, but otherwise there are no other lines within the map area of Figure 3c. The nearest seismic lines are sub-parallel to the line depicted, one lying 800 m to the east and another 2 km to the west. Balcombe-1 was mispositioned in Cuadrilla’s 2010 application by about 250 m to the north, as shown by the green arrow of its projection onto the seismic profile.

The blue line in Figure 3c shows the subsurface location of horizontal well Balcombe-2z. It was drilled ‘blind’ in a west-southwesterly direction, with no seismic control, only using measurement-while-drilling (MWD) technology at the drill bit. It ‘landed’ (deviated towards the horizontal) in the upper Kimmeridgian micrite (the I-micrite) and followed it for 757 m horizontally from the wellhead.

The application documents of 2010 included a map of PEDL244 in which the licence boundary of the excluded polygon, the ‘Bolney carve-out’, is in error by up to 1200 m. The existing Balcombe-1 well was also severely mispositioned, as stated above. These errors, due to careless mapping by Cuadrilla, should have led to a requirement to re-submit the application, but they were not recognised by either the MPA (West Sussex County Council; WSCC) or by the EA.

The Cuadrilla Balcombe history is an example of temporal salami-slicing of planning approval; first, get permission to drill and test the wells, and then seek permission to frack. There is no suggestion either in the first WSCC decision document granting a permit to drill, supported by the EA consultation, that either the council or the EA understood the implications of fracking. This is unsurprising, because Cuadrilla had avoided explicitly mentioning the technique. Nor did WSCC appreciate that horizontal drilling would extend beyond the limits of the approved site plan.

From the hydrogeological perspective the important issue about this appraisal drilling campaign is whether faults were recognised. The initial planning application ignored the existing BGS surface fault mapping; however, these faults have throws at the limits of, or smaller than, the vertical resolution of 2D seismic, which is about 30 m. The Paddockhurst Park Fault (PPF in Figure 3c) has a throw (displacement) of up to 30–40 m, but reducing to just 6–9 m SW of the Balcombe drill pad [87]. It is not recognisable on the seismic line; however, it is likely to have been transected by one or more of the Balcombe wells.

I have compared the Balcombe-1 released well log, as a proxy for Balcombe-2, with other well logs in the basin, as reproduced by the BGS [38], and have shown that the Paddockhurst Park Fault does indeed intersect Balcombe-1, and, by proxy, Balcombe-2/2z (for details see Appendix A: Cuadrilla at Balcombe). Cuadrilla has neither confirmed nor denied the existence of this fault, despite having had the opportunity to do so [88]. Cuadrilla’s evident ignorance of the faulting was ignored by the EA.

Cuadrilla at Balcombe also an example of a collaboration between academic researchers (in this case the University of Bristol) and a company in which the contractual link appears to have been hidden by the academic party. It is surprising that the Bristol team did not make transparent the collaboration with, and funding by, Cuadrilla. Cuadrilla was a sponsor of the BUMPS research consortium [89] (Phase 2) during the period of the research.

### 5.4. The Weald: Celtique Energie, Fernhurst and Wisborough Green 2014

Professor R. C. Selley was one of the eight members of RR2012, discussed above. He was asked by the South Downs National Park Authority (SDNPA, the MPA for the South Downs) on 24 June 2014 to comment on Celtique Energie’s Fernhurst application. He submitted his report [90] on 3 July 2014 (see Appendix A: Selley 2014 response to SDNPA letter of 24 June 2014). I in turn submitted a critique of his report [91], stating, in summary:

“Prof. Selley is complacent and uncritical regarding the completeness of the information supplied by the Applicant. He has failed to answer the question about whether the same geology can be found outside the licence area, choosing instead to offer some irrelevant information. He is factually in error regarding the Applicant’s targets. He is also inaccurate regarding the nature of the Kimmeridgian limestones, and again has failed to answer a specific question asked of him—whether or not the limestones will require fracking”.(see Appendix A: Smythe comments upon report to SDNPA by Prof. Selley)

While it was right and proper for the SDNPA to seek outside expert advice on the geological aspects of the application, and Professor Selley is an expert in the geology of the region, he nevertheless failed to provide accurate and essential information to the MPA.

Professor Peter Styles gave an invited presentation to the SDNPA about shale gas on 15 October 2013. He presented slides showing US-style simplified geology, with no faults present, and repeated the myth that more than 200 wells had been fracked already in the onshore UK. A fuller account of the failings of his presentation, which appeared to conclude that fracking in the UK would be safe, can be found in the Appendix A: Celtique Energie in the Weald.

I was asked by Lynchmere Parish Council to prepare a critique of Celtique’s application. I did so, and also submitted it in my own right as an objection [91]. In contrast to Professor Selley, I identified the following problems with the application:Misleading portrayal of the Kimmeridgian and Lias targets as conventional.Other Celtique information reveals targets as Bakken analogues (an unconventional US shale play).The illustrative seismic line for the site is 10–20 km distant, neither through nor adjacent to the site.This seismic line had been reprocessed, such that faults had become unidentifiable.

WSCC wrote to DECC and to the EA to ask for an opinion on my Wisborough Green submission. The EA stated that the majority of my concerns were around fracking. DECC responded:

“As he makes clear, his representations are entirely about the possible impacts of fraccing; since the application does not involve fraccing, we have no comments on his representations”.

This statement is untrue, in that many of my comments were about the erroneous or misleading geology presented by Celtique. The validity or otherwise of my observations remain equally valid for a conventional drilling programme. DECC also wrote to the EA and the HSE recommending that they read a blog article of September 2013 in which Dr James Verdon of Bristol University criticised my comments on Cuadrilla’s activities at Balcombe. The clear implication is given that DECC wished to have my views discounted because they came from an unreliable source; an indirect ad hominem attack.

The planning committee turned down the application in September 2014. It is not known how much weight, if any, was given by the SDNPA to the conflicting accounts of the geology supplied by the earth science experts.

### 5.5. Scotland Coal Bed Methane: Airth 2014

#### 5.5.1. Coal Bed Methane as a Cover for Shale Gas Operations

In 2013 Dart Energy was alleged to have been using its CBM plans at Airth [92] as a cover for shale gas operations in the same licence, PEDL133. The evidence included a statement by Dart to the Australian Stock Exchange (ASX) on its shale gas-in-place estimates [93], in which PEDL133 featured prominently. This evidence is important and likely to be accurate, because it is a criminal offence in Australia to make wrong or misleading disclosures to the ASX.

#### 5.5.2. SEPA Regulation Dependent on US Industry

The SEPA regulatory guidance on coal bed methane and shale gas [94] at the epoch of the inquiry refers to the US experience of fracking. Paragraph 30 states that SEPA controls impacts on the water environment using Controlled Activity Regulations [95] of 2011, and cites five heads which it regulates. None of these concern the nature of the geology to be drilled and exploited. The initial exploration licence (PEDL133 in this instance) was granted by DECC; likewise, the details of the geology were not examined during the licence examination.

Returning to the relevant CAR activities regulated by SEPA (omitting fluid injection and fracking operations, which were not part of Dart’s proposals), the geological or engineering aspect of SEPA’s regulation was limited to scrutinising the well construction, to ensure that the well did not contaminate groundwater. I exclude abstraction and treatment of flow-back water from this discussion, since that is essentially a hydrological, and not a hydrogeological, issue.

In the relatively new area of unconventional hydrocarbon exploitation (under which coal bed methane falls) there was thus no wider regulatory oversight of the potential environmental risks of drilling and stimulating vertical and/or horizontal boreholes.

SEPA’s guidance quotes a widely cited US oil industry paper, as follows:

“Figure 2 illustrates the maximum fracture length compared to typical aquifer thickness of the Woodford shales in the United States of America. This shows that the created hydraulic fractures remain within the planned range, even in the presence of faults”.

But the relevant figure below the text comprises a near-empty box including the statement “copyright SPE”. SPE is the Society of Petroleum Engineers. Below the empty box there is another diagram, presumably also referring to the Woodford Shale.

It is surprising that SEPA’s guidance should have depended on quoting, without adequate explanation, and without even an adequate citation, a copyrighted US oil industry paper [41]. This paper has to be purchased, and it is likely that few UK research institutions subscribe to the journal in which it was published. The paper was written by employees of a subsidiary company of Halliburton, a major hydrocarbon service company. The RR2012 report into shale gas development [40] accepted this same paper uncritically, as did a DECC report. This uncritical attitude towards an industry publication is surprising, given that:Halliburton has never published its database, which remains confidential.The paper was published in an industry journal; it is ‘grey’ literature, having been given only low-level peer review.Wells are only located by county, and individual wells cannot be identified.We do not know whether inconvenient results have been omitted.We do not know how complete is the database.There are no wells in areas where pre-existing faults reach up to the surface.

Halliburton emerged in 2015 as the third most corrupt industry group worldwide, as measured by the size of fines administered by the US department of justice and the Securities and Exchange Commission [96]. So why should we trust the views of such a company? However, even if we accept Halliburton’s main thesis at face value—that the creation of new fractures by fracking has a natural upward limit above the horizontal wellbore of around 500 m, perhaps 1000 m at the most—it is based entirely on US geology.

The UK shale basins are two to one hundred times smaller in area than their US counterparts, but their shale target is far thicker: typically 600–3000 m in the UK, as opposed to 30 m to 300 m in the case of the USA [97]. The US basins are mostly of foreland or intracratonic type, except for two extensional basins on the distal flanks of the Gulf of Mexico. The UK basins are of extensional origin, often developed during multiple discrete episodes, and sometimes with a local overprint of compression. They are cut by faults, which are predominantly normal, but sometimes re-activated compressively and/or by strike-slip. Many of the faults extend upwards from the shale to outcrop. In contrast, the US shale basins locally show occasional reverse faulting at the target shale depth, or minor extensional growth faults in the two Gulf plays, but it is extremely rare for any of these faults to extend up to outcrop.

Therefore, the claim by Halliburton that fugitive methane will not occur by passage up faults or artificially created fractures is inapplicable to the UK, and should not have been cited as valid evidence by the regulators and by the RR2012 expert committee.

#### 5.5.3. SEPA Allowing Discharges under Old Permits

A report from 2012, including video footage and based on Freedom of Information (FOI) requests [98], showed that SEPA continued to allow Dart to discharge waste water into the River Forth. Furthermore, Dart tested the composition of the water, not SEPA. The report stated:

“Dart say that the water is treated onsite before it is discharged into the Firth of Forth via their consented discharge point inherited from Hillfarm Coal Co Limited. This permission dates back to 1997. Dart inherited their permissions to abstract and dump water from Composite Energy, the company that previously owned the sites before new regulations came into force so no Hydro ecological (sic) assessment was done for the application. The permission allows them to dump 300,000 litres per day into the Firth of Forth. Permissions are being passed between companies, potentially avoiding updated legislation and making it hard for outside observers to find the information”.

The report also stated that SEPA had neither visited the site nor tested the waste water.

### 5.6. IGas Springs Road, Misson

The EA decision document summarised the 74 public responses, under the heading ‘Public responses relating to hydraulic fracturing’. The issues raised, in summary, were listed as:Contamination of groundwater, surface water and drinking water.Risk of earthquakes.Health impacts to local residents and wildlife.Nature of chemicals used in frack fluid.Object to government’s energy policy.Hydraulic fracturing contributes to climate change.

Under the heading ‘Summary of actions taken or show how this has been covered’ it explained:

“The application is for a mining waste permit to allow the management of waste produced from the exploration of mineral resources, in this case, from the drilling of two boreholes. The applicant has not applied to hydraulically fracture and as a result a permit will not allow the operator to conduct any hydraulic fracturing activities”.

This explanation is disingenuous, because my submission on the inadequacy of IGas’s geological understanding did not entirely depend on the presumption that fracking would take place. Nevertheless, the horizontal well drilled through the Bowland Shale was clearly an example of ‘gateway drilling’, in the hope that fracking would be permitted at a later stage. The chairman’s statement introducing the IGas financial statement [99] for 2019 described the “potentially world-class gas discovery at our Springs Road well site”. The report went on to make comparisons of the shale sequence with those of the USA such as the Eagle Ford, Barnett and the Marcellus, confirming its “suitability for effective hydraulic fracture stimulation”.

Another paragraph in the EA decision document summarised the concerns; “Hydrogeological report not adequate. Concerns were raised that geological fault lines had not been mapped correctly”, and answered it with:

“We have reviewed the Hydrogeological Risk Assessment (HRA) provided by the applicant against our information and conceptual understanding of the location. We are satisfied that the HRA has accurately documented the risks posed from the drilling of the two wells in relation to mapped fault lines throughout the targeted drilling zone”.

It appears that the EA has failed either to read in detail or to understand my submission. My submission as an objection, commissioned by Bassetlaw Against Fracking, pointed out that the application was the first stage of a progression towards fracking of the target Bowland Shale below the area. There could be no other possible outcome to this initial phase, apart from withdrawal from the site. Therefore, the application should have been considered in the context of the intention to frack. There would be no conceivable scientific nor economic merit in drilling the horizontal well per se, because the shale could not be exploited without fracking; this is an example of gateway drilling.

My evidence also highlighted the misleading nature of the geology as presented by IGas. Details are supplied in the Appendix A: IGas at Springs Road, Misson.

Lastly, my evidence also strongly suggested that the two search areas were selected on non-geological criteria, in which case the Applicant was misleading the Council.

### 5.7. Angus Energy, Brockham

#### 5.7.1. Appraisal or Production?

Angus Energy asked for a period of three years for appraisal. This is untenable, given that appraisal normally takes six months, usually less. Angus further stated in its March 2018 annual report:

“These activities from the Group’s conventional reservoirs at Brockham and Lidsey will be complemented by the testing of the Balcombe-2z well and the first commercial production from the Kimmeridge layers at Brockham (Brockham-X4Z) in 2018”.

It therefore appears that Angus was asking for an artificially extended period of ‘appraisal’, under cover of which it intended to start production.

#### 5.7.2. Conventional or Unconventional?

The appraisal of the Kimmeridge Clay Formation (KCF) was described ostensibly as a conventional resource appraisal. If the KCF were a conventional prospect it would comprise the following elements:The well-defined fault-bounded structure;The reservoir(s) of fractured micritic layers (semi-limestones or calcareous mudstones);The source of KCF shale below;The cap rock of KCF shale above.

However, this description is invalid, and, in any case, contradicted by Angus itself. Its Waste Management Plan of May 2017 stated (pp. 8–9):

“The new BRX4-Z sidetrack from the third well on the field offers the potential to produce from the Kimmeridge micrite limestones accessing oil from the Kimmeridge clay as a hybrid reservoir”.

The revealing phrase here is “hybrid reservoir”. Angus’s Investor Presentation of January 2018 defines hybrid reservoirs (slide 9) as “interbedded shale and limestone layers produced conventionally—without the need for fracking”. This definition is wrong and misleading. A hybrid reservoir is unconventional, although it may include elements of a conventional trap such as a sandstone or limestone. Examples of hybrid shale reservoirs include the Bakken play of North Dakota and the Niobrara of Colorado [100]. Both these plays require fracking to enhance the low permeability of the shale. The clastic and/or carbonate layers within the shale provide mechanically favourable targets for the fracking, but the bulk of the production comes from the fracked shale above and below.

In several of its published documents, including the one cited above, Angus discusses producing from the entire ~200 m thickness of KCF. It asserts that this entire section is “fractured”, and further asserts that fracking will not be required at Brockham. The proposed production from the entire 200 m thick section of KCF clearly conflicts with the definition of a conventional 30 m thick micrite reservoir prospect defined above, but, in contrast, compares closely to the unconventional (hybrid) Bakken and Niobrara plays of the USA.

The use of an acid wash only for wellbore cleaning is mentioned in the Waste Management Plan. There is no mention of matrix acidisation, a method of enhancing permeability in unconventional plays by injection of acid under pressure into the formation. Angus asserted that fracking (in the sense of high-volume hydraulic fracturing) would not be used. It is therefore difficult to see how Angus could ever achieve successful production from a geological sequence which elsewhere requires fracking and/or acidisation.

The contradictions in Angus’s plans have never been addressed by the EA.

#### 5.7.3. Was Brockham-X4z Drilled without Planning Permission?

The discussion herein is based largely on details provided by Brockham Oil Watch [101]. A timeline of communications between Surrey County Council (SCC) and Angus Energy, and site visit/meeting notes in the period leading up to and following the drilling of unauthorised sidetrack BRX4Z in January 2017 (see Appendix A: Angus Energy at Brockham) shows a change in messaging by Angus from planned drilling to a “maintenance workover”, following receipt of a letter from SCC in September 2016 stating that drilling was not permitted under existing planning consents. It also highlighted the confusion around well numbering created by Angus Energy. Angus told SCC on different occasions that it would be doing maintenance on Brockham-1 or well No. 2, but actually drilled from BRX4; there was also an urgent health and safety situation that was apparently engineered to disguise the drilling from SCC officers.

Following the drilling, Angus maintained that it had all required permissions for the drilling of sidetrack BRX4Z, disputing SCC’s position for months, and threatening the Council with legal action. Angus continues to maintain that sidetrack BRX4Z was fully authorised, but then applied for retrospective planning permission for its drilling.

A report [102] in March 2018 prepared by the planning development department of SCC stated:

“This unauthorised development has *highlighted discrepancies between the legislators of the oil and gas industry,* as permits for the drilling of a new sidetrack were issued by both the Environment Agency and the Oil and Gas Authority, and their legislative requirements do not require planning permission to be in place before they are issued. Whilst perfectly understandable in terms of legislation, it makes it somewhat confusing and at times misleading for both those involved and those monitoring such development” (my emphasis).

A fuller text of the report can be found in the Appendix A: Angus Energy at Brockham. It is noteworthy that the unauthorised drilling was witnessed by the local protectors’ camp, and might never otherwise have come to light.

#### 5.7.4. Brockham: Conclusions on Regulation

It can reasonably be concluded that Angus Energy has been mendacious and evasive about its plans and actions at Brockham. The MPA challenged the evident illegality of the sidetrack well, but then only requested Angus to submit an application for retrospective planning permission. It resisted Angus’s assertion that the existing well was authorised. Finally, on 8 August 2018, after delaying deliberation on this application three times (apparently due to insufficient information provided by the operator), the MPA’s Planning and Regulatory Committee granted permission, but without imposing any meaningful conditions in response to the main objections about the lack of an up-to-date environmental permit, the ongoing investigation into the Newdigate seismic activity (Section 4.13.2 and Section 5.9), and operator competence. The OGA maintained its habitual hands-off approach to disputes such as this.

The Angus Energy website [103] states:

“Further to our RNS of 28 June [2019], following further works undertaken in June, it is the view of the Company that, on any conventional approach, it is extremely unlikely that commercial hydrocarbon flow can be established from the Kimmeridge layer at Brockham. Nonetheless the Company is exploring all other options for the site including the resumption of extraction from the Portland reservoir”.

This statement appears to leave open the possible (albeit unlikely) future exploitation of the Kimmeridge Clay Formation by unconventional means.

The EA issued a variation to its original 2016 permit on 22 November 2018. Although the EA declined to enter the debate about the illegality of the BRX4Z sidetrack, the decision document [104] is unusually thorough, and concluded that re-injection of produced water back into the Portland Sandstone Formation (the producing layer) via the re-injection well BRX3 would not be re-permitted. Its refusal was based largely on the lack of provision of satisfactory information by Angus. There were no satisfactory logs of the well (drilled by the then operator Midmar in 2007) to prove its integrity, nor was any relevant information held by HSE.

Angus failed to demonstrate that it had the proper procedures in place for monitoring of well integrity, which the EA had requested in the absence of any proposed groundwater monitoring wells or information on the cementing or integrity of the ‘as installed’ well. The Water Injection Procedure for Brockham was described by the EA as “newly developed” so it appears that Angus had no procedures, either for reinjection or for monitoring of well integrity, until it was asked about them.

Acid washing of the well would be permitted, subject to the provision of adequate information in advance. No production would be permitted from the KCF until the EA approved an extra pre-operational condition concerning gas management. The EA considered the permit re-application to be of high public interest. The strong and well-informed consultation responses from Brockham Oil Watch and other local action groups undoubtedly ensured that in this example the permit determination was thorough, and not merely a box-ticking exercise.

Lastly, the inability of Angus to provide records of the cement bond log for the re-injection well illustrates the danger of sites like Brockham passing through the hands of many different operators (five in this case) without adequate records being kept. This is a failure by the HSE; firstly, to require such logs and measurements to be furnished by the operators on a regular basis, and secondly, to archive them for posterity. Preese Hall-1 and Horse Hill-1 are other examples of similar failure of record-keeping by the HSE.

### 5.8. UKOG Broadford Bridge

The PEDL234 licence was originally awarded to Celtique Energie in partnership with Magellan Petroleum on 1 September 2008. The Initial Term Work Programme comprised a requirement to acquire 150 km of 2D seismic, and to reprocess 300 km of existing 2D seismic. There was a one-well drill-or-drop clause. The OGA asserted [105] that, after almost eight years into the licence term, the construction of the Broadford Bridge well pad was evidence of “substantial progress” (see Appendix A: OGA response to Sutcliffe letter 7 July 2016). The OGA granted the new operator UKOG a two-year extension to the PEDL234 licence, even though the deadline for a licence term extension had passed on 31 May 2016, and the previous operator had been given two extensions.

The OGA permit issued on 25 May 2017 was based on UKOG’s revised plans for testing the ‘limestones’ of the Kimmeridge Clay Formation, but, nevertheless, the permit issued by WSCC for the original conventional Sherwood Sandstone Formation target was never modified, nor was a new permit sought from the MPA (more details are given in Appendix A: UKOG at Broadford Bridge).

UKOG submitted a request for an extension to planning approval in a statement [106] dated 6 July 2017. It was mainly an appeal for an extension of twelve months to fulfil the terms of the licence. Under the heading ‘Application Detail: Full Planning Permissions consented MPA ref: WSCC/052/12/wc’ it states:

“The siting and development of a temporary borehole, well site compound and access road including all ancillary infrastructure and equipment, on land at Wood Barn Farm, Broadford Bridge, for the exploration, testing and evaluation of hydrocarbons”.

However, the full permit description qualifies the statement above, to read:

“… for the exploration, testing and evaluation of hydrocarbons *in the willow prospect*” (emphasis added).

UKOG wilfully omitted this crucial condition regarding where it was permitted to drill to. In addition, there is no mention in the planning statement of the Kimmeridge Clay Formation. UKOG then went on to drill a deviated well to target the Kimmeridge Clay Formation, but ran into the severe technical problems that I had predicted would occur in my EA objection submission [107] of April 2017.

UKOG ran an aggressive smear campaign to try to denigrate my evidence, as well as that of Mr. Graham Warren, a retired EA hydrogeologist.

In summary, Broadford Bridge provides evidence of severe regulatory inadequacies comprising:Continuing and unjustified licence extensions by OGA.Operator drilling before EA permit in place, but no sanction applied.Operator not drilling according to the MPA planning permission, but no sanction applied.Operator using a prior approved conventional drilling plan (the Celtique application of July 2012) to test the Kimmeridge unconventional play.Operator misleading the MPA as to the purpose of the amended drilling proposal of July 2017.

The OGA and the MPA should have cross-checked with each other that they were both approving the same work programme. It is evident that a new MPA permit should have been applied for, because, inter alia, the justification of the drilling site chosen originally by Celtique Energie became invalid with the change of target from Triassic sandstones (in a conventional geological structure) to ‘limestones’ within the KCF. Local objectors did point all this out in a letter to the MPA, but the MPA dismissed their objections, on the ground that the change of target concerned hydrocarbon development, and was therefore beyond its planning remit.

The current PEDL licence still includes the obligation to shoot 150 km of 2D seismic data. Celtique Energie shot 170 km of 2D data in 2011 in Surrey and Sussex, but only 33 km of this was inside PEDL234. So Celtique never fulfilled its licence obligations, which then passed to UKOG. To date UKOG has shot no seismic surveys in the onshore UK, according to the UKOGL [18].

In addition, the drilling of Broadford Bridge-1 and its sidetrack were both severe technical failures, involving loss of well integrity. No remedial measures have been required by the regulators.

### 5.9. UKOG Horse Hill

Horse Hill-1 became known as the ‘Gatwick Gusher’ [108] after claims by UKOG in February 2015 that the Kimmeridge Clay Formation was a world-class source of oil.

The Officer Report for the SCC planning meeting of 11 September 2019, recommending approval of the development, failed to address various problems, including the possible link of drilling to the new earthquake activity, the proximity to a fault, the use of acidisation to stimulate the KCF, and the many errors in the scanty cartoon-standard geological interpretations by UKOG (see Appendix A: UKOG at Horse Hill). At the planning council meeting which approved the development, the deputy planning development manager assured councillors that the county council was obliged to rely on the finding of the OGA that the earthquakes were “natural” (not human-induced) [109]. 

The OGA report of a workshop [80,110] hosted by the OGA on 3 October 2018 concluded by a majority that drilling or any subsurface activity at Brockham was very unlikely to have been a cause of the earthquake swarm. It discounted the hypothesis presented by Edinburgh University researchers [111] that surface operations at Horse Hill may have had an almost immediate physical effect in the subsurface, namely, the release of the wellhead pressure by opening of a valve on, or just prior to, the first earthquake of 1 April 2018. According to the operator, the HSE holds records of all well integrity tests carried out between 2016 and 2018, spanning the period when the earthquake swarm began. But the HSE has denied holding these documents. In either case the HSE has powers to demand that the operator supplies the HSE with another copy, but it has not done so. This is a basic failure of regulation by the HSE concerning the important matter of whether the earthquake swarm was human-induced or not.

### 5.10. Lancashire 2009–2013

#### 5.10.1. Hale Hall

The Hale Hall planning application shows that Cuadrilla was engaged in unconventional exploration in Lancashire right from the start in 2009 (see Appendix A: Lancashire 2009–2012). In contrast to its exploration in other regions, Cuadrilla has never tried to disguise the fact that its exploration in Lancashire was unconventional.

#### 5.10.2. Grange Hill

The well prognosis for Grange Hill-1 included Mercia Mudstone Group (MMG) at outcrop, wth the top Sherwood Sandstone Group (SSG) at about 110 ms of two-way reflection time (c. 90 m) below sea level. No comments were made by the EA about the inadequacy of such confinement of the SSG. In the event, the well spudded straight into unconfined SSG (see Figure 6). It is reprehensible and surprising that the EA did not see fit to comment on the risk to this Principal Aquifer at or near the surface.

My independent mapping of the area around Grange Hill-1/1z using the now-available 3D seismic survey shows that there are several faults trending NE-SW through the well zone. Three of these cut the well obliquely at steep angles, at depths from the Crossdale Mudstone down to near the base of the Sabden Shale. Their existence explains the severe problems that the operator encountered.

If the 3D survey had been acquired and interpreted before the drilling of Grange Hill-1/1z then the faults would have been recognised and the problem avoided. However, the well was sited using a 2D E-W seismic line dating from 1979, reprocessed but still of poor quality, and situated some 800 m to the south. This was a completely inadequate image of the subsurface on which to plan any drilling in the twenty-first century, be it conventional or unconventional.

A scoping report prior to a full Environmental Impact Assessment (EIA) was submitted to LCC by the operator on 9 July 2013. A planning application was submitted on 22 May 2014 for seismic and pressure testing, but it was argued that a full EIA was not necessary. No updated geological interpretation was provided with the application, although by then the operator would have been in possession of the 3D survey for well over a year. The BGS was asked to comment on the scoping report, but declined, writing [112]:

“I am afraid that BGS is not in a position to comment meaningfully on the planning application. The geology summary presented in the application seems reasonable. We are unable to comment on the plans that Cuadrilla propose for the well”.(see Appendix A: BGS comments 2013 to scoping opinion, Grange Rd)

The geological summary referred to above was no more than a table of formations encountered at the well. The BGS lost the opportunity here to comment on the existence of the 3D survey (a copy of which would have been in its possession) and to make pertinent comments about the well.

Grange Hill was formally abandoned when the development application was withdrawn on 14 November 2014. But in February 2016 Cuadrilla won an appeal against refusal of the use of the site for seismic and pressure testing. However, this permission was never acted upon.

The Grange Hill history can be categorised as a combination of technical failure and deception by the operator.

#### 5.10.3. Anna’s Road-1

Anna’s Road-1 was abandoned on 4 October 2013 due to “technical constraints related to wintering birds”, according to the CEO of Cuadrilla, who stated that their planning permission only allowed them to drill for six months of the year, and that this did not allow them to develop the site. This decision might appear to be an example of strong regulation in practice, but it is spurious. In fact, a tool for testing the cement bonding had become stuck in the open hole at about 600 m depth in the SSG the previous October. While it is correct that drilling was not permitted during the winter, that delay would not have prevented Cuadrilla restarting work in the spring of 2013. The Anna’s Road history is best categorised, once again, as a combination of technical failure and deception by the operator.

#### 5.10.4. Preese Hall-1

Preese Hall-1 was noteworthy for being the first UK shale well to be fracked, as noted above [113]. Given its importance as the first well of a potential new era of unconventional exploration in the UK, the sequence of work (see Appendix A: Lancashire 2009–2012) shows that the well was drilled and fracked prematurely. UK regulatory guidance on requirements for operators, published in 2013, now expects that the acquisition, processing and interpretation of the 3D seismic survey, and the installation of a network of local seismometers, should all be carried out *before* a well is tested [114].

The planning application [115] stated:

“The Bowland Shale gas reservoirs in these wells (Preese Hall-1 and Thistleton-1) are described as ‘unconventional’ reservoirs. In such reservoirs the source, reservoir and seal rocks are all in the same formation and methane gas in widely distributed as adsorbed and free gas throughout the formation in relatively low concentrations. Subsurface stimulation is needed to help flow the gas to surface”.

No application appendices are available on the LCC website. Only one consultation response was received from statutory consultees, from Fylde Borough Council. The planning application delegated report of 30 October 2009 stated that no response had been received from the EA. So, it appears that the MPA approved the application without understanding the implications of ‘stimulation’ of an unconventional resource. The fault here lies with the EA for not realising the implications of what Cuadrilla explicitly said in its application was “a shale gas (well), one of the first of its kind in Europe”.

Charles Hendry, the Minister of State at DECC, wrote to Lord Browne, Chairman of Cuadrilla Resources Ltd., the operator, on 11 May 2012 to express his concern that the wellbore deformation, which might be linked to the fracking, had been concealed from his officials [116] (see Appendix A: Charles Hendry (DECC) - Cuadrilla correspondence 2012). This well casing deformation is crucial to understanding the tectonics. Mr. Hendry stated:

“You will be aware that my Department is concerned that Cuadrilla failed to recognise the significance of the casing deformation experienced in the earth tremor triggered by tracking operations on 1 April 2011. So much so, that the company did not report it to my officials in contemporary discussions as to the possible cause of the tremor and the possibility that it might be linked to the fracking. In the light of Cuadrilla’s responses to the Department’s subsequent inquiries, I have formed the view that this failure discloses weaknesses in Cuadrilla’s performance as a licensee, which need to be addressed”.

This was a damning indictment by government of Cuadrilla’s incompetence. Nevertheless, no sanctions were applied, and Cuadrilla has continued to date as operator on several UK PEDLs.

There were other safety issues concerning Preese Hall-1. Firstly, the well integrity; the high annular pressure (300 psi; this is double the threshold at which a loss of well integrity is indicated in Canada and the US [117]) was allegedly “mitigated”. Permission was given to plug and abandon this well, with no ongoing monitoring. Secondly, a small quantity of flowback water from the well containing high levels of radioactive elements was dumped in the Manchester Ship Canal, with permission to do so being retrospective (see Section 5.12.2 below).

The EA’s view on the seismicity and consequent wellbore deformation of Preese Hall-1 is illustrated by an email exchange between the EA and a local resident, dated 3 October 2014. It stated:

“The seismic activity caused a deformation of the borehole close to the base of the well around 8500ft below the ground and the independent report that Cuadrilla commissioned for the Department of Energy and Climate Change (DEC) [sic] ‘Geomechanical study of Bowland Shale Seismicity’ dated November 2011. The location, at greater depth than the higher perforated and hydraulically fractured zones and degree of the deformation makes it clear that the well bore integrity was not compromised”.

This statement reveals a severe lack of geological understanding by the EA. The complete exchange between the EA and the resident is reproduced in the Appendix A: Correspondence between the EA and local resident 2014. The wellbore deformation was so severe that the hole had to be abandoned. The well integrity was also compromised, because the cement bond around the pipe would have been cracked. The EA went on to say that it was satisfied that an environmental permit had not been required, therefore that the well could henceforth be plugged and abandoned “in line with the requirements of the Health and Safety Executive”. The resident correctly pointed out that in future neither the HSE nor the EA would be monitoring leakage from an evidently badly damaged well.

Cuadrilla and co-authors published a paper [78] purporting to show that the fault on which the main seismicity took place did not intersect the wellbore. I demonstrated the inadequacy of this conclusion in a discussion paper [97], showing that a more realistic interpretation of the published seismic section through the well (taken from the 3D data volume) puts the seismically triggered fault intersecting the well in the precise zone where the wellbore is deformed. The Cuadrilla reply [88] to this part of my discussion was unconvincing. My interpretation of the full 3D seismic dataset, accessed in late 2018 after its delayed release, confirms my re-interpretation of the fault as intersecting the borehole, although the faulting is actually somewhat more complex than previously depicted either by myself or by Cuadrilla in 2016.

In conclusion, the view of the EA that the seismic fault did not intersect the wellbore, and that the well casing deformation was not serious, is both complacent, and also misguided, because it is not evidence-based.

### 5.11. Europa Leith Hill

The choice of drill site and the oblique Z-shaped nature of the drilling required to reach the Holmwood prospect to the south raised many questions (see Appendix A: Europa at Leith Hill), not least the existence of a drill pad in the Forestry Commission woodland, itself situated in the Surrey Hills Area of Outstanding Natural Beauty. Local residents had been fighting the plans since 2008, including two Public Inquiries, a High Court Appeal and a Court of Appeal Hearing. The second Public Inquiry found in favour of Europa, but the company was never able to meet all the conditions of its planning permissions, such as producing an acceptable Traffic Management Plan.

The concern discussed here is the risk to the Hythe Formation, a principal aquifer. This aquifer is unconfined; that is, it is at the surface at the wellsite. Sutton and East Surrey Water wrote to the MPA in December 2014, registering a formal objection to Europa’s proposed development on the ground that this aquifer might not be protected in the event of a leak or a spill. Envireau Water prepared a hydrogeological risk assessment for Europa in March 2015. It purported to show that there was no hydraulic connection from the drill site to the important Dorking water supply boreholes some 5 km to the north. However, in my submission [118] to the EA consultation of early 2018 I showed that there were serious errors in the Envireau geological interpretation and its resulting conclusion (details are provided in the Appendix A: Europa at Leith Hill). I showed that there was indeed hydraulic continuity from the wellsite down-dip to the north, via the Hythe Formation itself and permeable superficial deposits, directly to the Dorking water boreholes. In response the EA stated in its permit award [119]:

(Heading: Response concerns) “Concern over poor understanding of the structure of the shallow geology of the Hythe Formation Principal Aquifer and misleading conclusions on the groundwater flow direction. …

(to which it responded with)

The Environment Agency is satisfied that adequate information has been submitted in the Site Condition Report, the Hydrogeological Risk Assessment and the Technical Note to show how the groundwater immediately beneath the site, the groundwater that feeds to the local Pipp Brook and that provides water to local springs, private supplies and, indirectly, to public water supplies, would be protected. We have considered the risk to the Hythe Formation Principal Aquifer and concluded that the management practices set out by the applicant are sufficient to mitigate the risk”.

The EA refused to consider the risk to the Dorking public water boreholes, falling back on the supposed adequacy of Europa’s plans to “mitigate the risk”, despite its evidently inadequate and out-of-date geological interpretation.

The EA also failed to see that the proposed conductor casing was too short, at 50 m, to protect the Hythe Formation and the underlying Atherfield Formation, simply reiterating that it was:

“satisfied with the proposals to install the conductor casing into the top of the Weald Clay, at approximately 50 m true vertical depth”.

Other issues were not adequately taken into account by the EA. In conclusion, the EA’s understanding of the geology, based upon what was presented to it in my consultation submission, is severely limited. Overall, the permit uses the phrase “we are satisfied” fourteen times, and ‘satisfactory’ is also used in a positive sense eleven times. In my view this implies a considerable degree of complacency regarding the scrutiny of the proposals.

Had the drilling proceeded, it is probable that the same severe technical problems would have arisen as at Broadford Bridge (see Section 4.7.3 above), due to the very shallow angle of the deviated well when transecting faults. Such a possibility was also ignored by the EA.

### 5.12. Lancashire 2013 to Date

#### 5.12.1. IGas Irlam-1 and Irlam-1z, 2014

The two wells had explicit permission for coal bed methane exploration to a depth of 1372 m, but were drilled much deeper, to 2240 m and 2151 m (true vertical depth below sea level) respectively, terminating in Lower Carboniferous Pendleside Limestone. The wells were drilled deeper than permitted by the planning permit to test the Bowland Shale. Salford City Council (the MPA) responded to an FOI request [120] questioning the legality of this as follows:

“We can confirm this was a *minimum* and not a maximum depth. There is no maximum depth for drilling attached to the planning permission” (emphasis added).

Even if correct, this response begs the question of why the target of coal bed methane was even mentioned in the planning permit, if an operator can feel free to disregard it. It also implies that if an operator fails to reach the specified depth then it will be in breach; this is not a credible legal position to hold, since wells frequently fail to reach the intended depth. The response went on to state that IGas had submitted plans to the EA for “this different operation”, implying that the details of the geology were beyond the remit of the MPA. Note that this FOI request and response were dated November 2013, shortly before the first well Irlam-1 was spudded on 10 January 2014.

#### 5.12.2. Flowback Disposal

There has been controversy about untreated naturally occurring radioactive material (NORM) from the Preese Hall-1 drilling operation being dumped in the Manchester Ship Canal. My understanding of this is as follows. After treatment at the Davyhulme water treatment plant, some 3850 m^3^ of flowback water from the well was discharged into the canal. However, it had not been treated for NORM. A further 81 m^3^ remained behind in tanks at the drill site. After about a year the EA permitted Cuadrilla to ‘dispose’ of this volume by sending it to test centres such as Remsol for ‘testing’. It was radioactive to an average of 90 Bq/l, whereas the permitted level for disposal is 1 Bq/l. Was this testing ploy a means of bypassing regulations? A problem is that during the storage period the EA regulations changed. This history illustrates that either Cuadrilla and/or the EA have acted in a non-transparent manner.

#### 5.12.3. Mercia Mudstone Group as a Seal

The EA granted a permit to Cuadrilla for Roseacre Wood in February 2015. The geology of the area in question potentially has all three ingredients for a successful conventional hydrocarbon play:Source—Bowland Shales.Reservoir—Sherwood Sandstone Group.Seal—Mercia Mudstone Group.

The Fylde lies, geologically, within the East Irish Sea Basin, where oil exploration has been more successful than in the Central Irish Sea basin further south. This is attributed [121] to the presence of ductile impermeable halites (salt rock) within the Mercia Mudstone Group (MMG); the halites re-establish seal integrity after a period of inversion uplift, during which these shales were taken into the brittle tensional strength field.

How much rock is required above a fracked zone to seal it? In the Central Irish Sea Basin it was concluded that at least 600 m of MMG is required for it to be an effective hydrocarbon seal there, due to the inversion uplift, and in the absence of the ductile and sealing halites that are present in the East Irish Sea basin. 

In the Fylde the MMG is generally only 200 to 300 m thick, and in the Grange Hill-1 area it thins out to zero, leaving unconfined SSG at outcrop below Quaternary (Figure 6). Halites are absent in the target area, although they were once present, as proven by collapse breccias and pseudomorphs after halites. Halites are present further west north of Blackpool, at the Preesall Salt Field [122]. Therefore, the MMG, if depleted of its original halite layers, is proven by the Irish Sea exploration history discussed above to be completely inadequate as a seal to prevent upward migration of liquids and gas.

#### 5.12.4. Confined Sherwood Sandstone Group as an Aquifer

The EA has written off all the groundwater in the Sherwood Sandstone Group (SSG) below the Fylde west of the Woodsfold Fault as being saline, static, and therefore not of concern, should it be contaminated by drilling and fracking activities. Figure 8 shows a schematic cross-section running east–west across the area of Figure 6. The principal aquifer of the SSG is at outcrop (below Quaternary) east of the Woodsfold Fault. The EA accepts that the aquifer is recharged by rainwater mainly falling on the Bowland Fells, to the east, and that it flows across faults there. However, the EA has modelled the major Woodsfold Fault as a non-transmissive flow boundary, and has therefore written off the confined groundwater resource of the SSG west of this fault, despite evidence to the contrary that the groundwater in the SSG here is potable and at shallow depth.

My view is that there is good evidence for fresh water abstraction west of the fault, and that the inferred flow pattern is west across the fault as shown by the dotted blue arrows in Figure 8. In addition, it is possible (but not proven) that the Wakepark Fault, which crops out just west of Preston New Road, may feed the Wakepark Lake.

Details are provided in the Appendix A: Sherwood Sandstone aquifer below the western Fylde. In summary:The location of the Woodsfold Fault was uncertain to within 1 km in an east-west direction (Figure 6); my recent re-interpretation of the 3D seismic data volume shows that it comprises two separate faults, the Woodsfold Fault and the Hesketh Fault (Figure 6).The EA is unjustified in assuming that the Woodsfold Fault is a non-transmissive boundary. It cites the modelling studies of 1997 and 2009, in which the fault was *defined* as a boundary, a priori. Therefore, it cannot then take the modelling results as proof that the fault is a barrier to flow, even if the seasonal and annual modelling results satisfactorily explain the measured data east of the fault. A sensitivity study is required to model the transmissivity of the fault.The Kirkham borehole, which encountered hypersaline water, penetrated salt horizons within the Mercia Mudstone Group. Use of this anomalous result cannot then be applied to the whole of the western Fylde. The same applies to the recent deep borehole drilled by the BGS near Roseacre Wood. Figure 6 shows that both The Kirkham and the BGS Roseacre boreholes are situated in an anomalously deep portion of the top-SSG structure, in a poorly-defined graben or half-graben trending north-south (Figure 6).The top SSG (base of the Mercia Mudstone Group) is shallower than 200 m depth over approximately half of the map area shown in Figure 6; water in this confined SSG aquifer at such shallow depth is likely to be fresh, as implied by several water boreholes which penetrated to the SSG in search of water, and shown schematically in Figure 7; the EA is unaware of the implications of two historic boreholes which penetrated to the SSG, and which presumably drew fresh water (for more details see Appendix A: Sherwood Sandstone aquifer below the western Fylde).Elsewhere, in the East Midlands, the SSG passes downdip eastwards to be confined below the MMG [123], within which the groundwater remains fresh down to about 500 m depth [124].The EA has never sampled, nor considered the implications of, the water in the Wakepark Lake (Figure 6); it asserts without justification that the lake is recharged year-round by a local sand lens within the Quaternary.The Wakepark Fault, if acting as a conduit for upward flow from the confined Sherwood Sandstone aquifer, completes the discharge end of the cycle of recharge-discharge, which was missing from the EA’s investigations, and which the EA uses as an argument for no flow below the western Fylde.

In conclusion, the dismissal by the EA of the entire confined SSG aquifer west of the Woodsfold Fault as saline, and therefore of no potable value, is both wrong and reprehensible. Even if little-used at present, it may become a necessary water resource in the future. In addition, the possible contamination of the nationally important Primary Aquifer east of the fault, however remote the possibility, should have been more carefully considered. The bounding Woodsfold Fault needs both accurate remapping (as I have done within the south-east corner of the 3D survey – see Figure 6) and a new hydrogeological study.

#### 5.12.5. Manchester Marls as a Seal

A crucial facet of Cuadrilla’s claim that no fracking-related contamination will travel upwards at either of the two sites is that the Permian Manchester Marls underlying the SSG will provide the required seal. This layer is shown by the uppermost half of the pink layer in Figure 7, the lower half being the Collyhurst Sandstone. However, this layer is only 270 m thick in the area of the proposed Roseacre-1 well [125]. Cuadrilla stated:

“The Manchester Marl locally forms a seal to underlying hydrocarbon bearing geological units. The Collyhurst Sandstone is the gas reservoir at Elswick gas field in central Fylde, where it immediately underlies the Manchester Marl”.(the location of Elswick-1 is indicated in Figure 8)

This statement is misleading because it implies that the sandstone is a conventional gas reservoir requiring a top-seal. However, the Collyhurst Sandstone is a low permeability, low porosity sandstone, which required fracking to produce the gas (although it should be noted that conventional fracking of this vertical conventional well [126], using gelled water with CO_2_, has little in common with high-volume horizontal fracking of shale). Therefore, its former modest success as a gas producer owes little or nothing to the overlying Manchester Marls acting as a seal. A fuller discussion is provided in Appendix A: Manchester Marls as a seal.

The thickness variation of the Manchester Marls across faults, as illustrated schematically in Figure 8, is attributed to syndepositional movement on the faults [123]. The Manchester Marls have a similar lithology and origin to the Mercia Mudstone Group. Therefore, their mechanical and hydrogeological characteristics will also be similar, and it is unlikely that they will provide an adequate seal to prevent upward fluid migration. In addition, the layer is cut by numerous faults, many presumed to be syndepositional in nature. Such faults zones will comprise breccias and crush rock, and so are unlikely to be good fluid seals; on the contrary, they will be transmissive to fluids.

In conclusion, the Collyhurst Sandstone and overlying Manchester Marls, whether faulted or not, are an unreliable seal for upwardly escaping fluids. The EA appears to have accepted uncritically the misleading geological views put forward by Cuadrilla.

#### 5.12.6. Downhole Seismic Monitoring Array

I submitted an objection [127] to the EA consultation (see Appendix A: Smythe objection 2017 to PNR variation of permit for microseismic monitoring) to the request by Cuadrilla to alter the originally approved seismic monitoring array, which comprised ten stations at the surface and 80 stations buried just below the surface. The ten stations would be retained, but the 80 stations buried at 100 m depth were to be replaced by a downhole string. I argued that the downhole string was a technically inferior replacement for the 80 buried stations, especially for one of its main objectives, the location of seismic events triggered on pre-existing faults. My objection was ignored.

In the event my fears about the efficacy of the downhole string were justified by the monitoring that took place between October and December 2018 when fracking was being conducted in PNR-1z. Whereas the locations of the microseismicity were well constrained by the downhole array, the estimation of fault plane solutions for the larger events was extremely poor. Clarke et al. [128], in a paper co-authored by Cuadrilla and University of Bristol employees, admitted that “many of these larger events produced subsurface motions that were beyond the dynamic range of the downhole instruments”, so that accurate magnitudes M_w_ could not be determined. The phrase “beyond the dynamic range” means that the seismometers were physically incapable of recording large signals. This is an astonishing failure by the geophysicists responsible for the design of the arrays. Furthermore, due to this design failure, fault plane solutions could be estimated for only six events out of the 69 recorded events with M_w_ > 0.0.

In conclusion, the EA should not have permitted Cuadrilla to degrade the quality of the initially approved seismic monitoring arrays.

#### 5.12.7. Release of the Bowland-12 3D Seismic Survey

The Cuadrilla 3D seismic survey that was conducted in the Fylde in 2012. It covers the area within the dashed black line shown in Figure 6. According to the OGA’s Consolidated Onshore Guidance, such surveys are required to be archived and made available to the public after a given period of time. The survey information is passed to the UKOGL, which initially publishes the location of the survey and, after the expiry of the relevant confidentiality period, publishes the full survey information on demand [33]. In this instance the survey fell due to be released by the UKOGL on 1 January 2018.

I corresponded with the UKOGL to ensure that I could obtain the dataset in early 2018. The data were prepared and made ready for supply, but in March 2018 UKOGL informed me that the dataset had not yet been authorised for release. An FOI request to the OGA elicited this response:

“A formal representation to extend the confidentiality period for the Bowland-12 3D seismic survey was made by Cuadrilla Resources on the 9th January 2018 following a verbal representation on the 3rd January 2018. The Oil & Gas Authority are currently evaluating the representation”.

Furthermore, the OGA treated the FOI request as “ordinary business”, because my request was “not a request for information actually held by the OGA”. Although, to the best of my knowledge, there is no precedent for the withholding of data from release, might be argued that it was being done to protect Cuadrilla’s commercial interests. However, the data have no intrinsic commercial value, because Cuadrilla has an exclusive licence to the area until 2031. The geological knowledge gained from the data within the licence area has no significant value outside the licence. The only value in the data for outsiders is to enable scrutiny of Cuadrilla’s work. It appeared that Cuadrilla, with the connivance of the OGA, was seeking to avoid such scrutiny.

#### 5.12.8. Hydraulic Fracture Plans

Cuadrilla submitted its first hydraulic fracture plan (HFP) for PNR-1z in November 2017, as required by the EA.

Preston New Road Action Group (PNRAG), with myself acting as its geological expert, challenged the validity of Cuadrilla’s PNR-2 HFP in November 2018, via Harrison Grant Solicitors. My report for PNRAG was also submitted as a formal objection to the EA consultation process [129]. The EA did not respond other than by acknowledging receipt. Estelle Dehon, now acting for PNRAG, wrote to the EA on 12 July 2019, highlighting additional shortcomings in Cuadrilla’s HFP for PNR-2, and stating:

“there remain serious and fundamental errors in the subsurface geological interpretation reflected in Cuadrilla’s revised draft HFP which should, unless corrected, lead to the EA refusing to approve the HFP”.

The EA responded substantively [130] only on 22 July 2019 to the legal letters of November 2018 and July 2019 (see Appendix A: EA response to Estelle Dehon 22 July 2019), and approved the HFP a few days later, rejecting the arguments that I had put forward. The EA’s response is analysed next.

Figure 9 shows the development of the geological interpretation by Cuadrilla along an E-W profile through the PNR wells, over a period of nine months from November 2017 to August 2018. A subsequent version used in the finally approved version of the HFP, dated June 2019, differs only in trivial detail from the version shown in Figure 9c.

Figure 9a shows that Cuadrilla prognosed some 300 m of Millstone Grit below the Collyhurst Sandstone, but in January 2018 the PNR-1 drill bit passed directly from the Collyhurst Sandstone into Upper Bowland Shale; the Millstone Grit was entirely absent. The EA claimed to have asked Cuadrilla to modify its geological interpretation. Cuadrilla did so by ‘photoshopping’ the coloured cross-section images as shown in Figure 9. Firstly, it turned up the base of the Millstone Grit west of the wellbore to ‘solve’ the problem of the missing rock (blue ellipse in Figure 9b). Then, because this made the Bowland Shale structure to the east look unrealistic, it altered the image again to even out the thicknesses of the Upper and Lower Bowland Shale layers–albeit now tilted at up to 80° (the blue ellipse in Figure 9c). None of these changes were based on an actual reinterpretation of the seismic data volume, as should have been done; the seismic data shown in the HFPs clearly contradict the photoshopping attempt at ‘curing’ the problem. The EA responded to an FOI request (CL101888HR) concerning these anomalies as follows:

“The operator still believes this formation to be present, from the interpretation elsewhere. As a result they have retained the original interpreted level of the top Bowland shale where it is above the lateral to be hydraulically fractured. This interpretation has been the subject of discussion between the Oil and Gas Authority (OGA), ourselves, and the British Geological Survey.We have accepted the interpretation in the HFP on the basis that the alternative is that the top Bowland shale horizon corresponds to a higher reflection event, and as this horizon defines the upper permit boundary, the current interpretation of a lower horizon is a safe one”.

Therefore, the EA accepted the scanty and evidently specious geological cross-sections and explanation given in the latest version of the HFP, based on vague assurances (“the operator still believes *...*”) and discussions with other agencies (OGA and BGS). This complacent attitude is inconsistent with sound regulation.

#### 5.12.9. Wakepark Fault and Wakepark Lake

The EA’s response [130] to the omission of the Wakepark Fault (Figure 6 and Figure 8) from Cuadrilla submissions states:

“The presence of a fault which is actually poorly defined in the 3D seismic data volume has been reviewed together with the hydrogeological evidence in the area of the Wakepark Lake. … So, despite the seismic potential of a fault, it is clear from the hydrogeological evidence that even if the fault is present, and it did link the sandstone with the surface, it is not conducting water upwards to the Wakepark Lake”.

It appears that the EA is reluctant even to admit the existence of this fault, which was first identified by industry exploration 30 years ago. The fault is not “poorly defined”; it is about 4 km long, with a throw of up to 100 m. It comes up to outcrop above the toes of the two PNR lateral wellbores. But Cuadrilla, and, for that matter, the OGA and the BGS (EA consultees on the question), have failed to take account of it. Such failure is simply due to poor interpretation of the data by an incompetent operator and by the public agencies; the allegedly poor definition of the fault in the dataset is untrue, and in any case would be an untenable explanation for its omission.

The EA responded to an FOI request quoted above about the 3D survey interpretation failings by Cuadrilla as follows:

“We do not assist in the original interpretation or re-interpret the operator’s data volume, but in this case the 3D dataset has been reviewed by the OGA. Please note that a complete reinterpretation of such a 3D seismic data volume could potentially take weeks”.

This response shows that the EA seems to have been more concerned about saving Cuadrilla time and money than about sound regulation. However, Cuadrilla had had some eighteen months since the unexpected results of PNR-1 came in, and therefore had ample time to fully re-interpret the 3D survey. I was able to solve the problem of the missing Millstone Grit and remap it within a week or so of obtaining the 3D dataset [129]. So Cuadrilla could (and should) have done the necessary re-interpretation by early 2018, within a few weeks of receiving the surprising PNR-1 results. It is unacceptable that the EA and the OGA were prepared to accept photoshopped pseudo-interpretations by Cuadrilla, instead of insisting on a proper re-evaluation of the whole data volume around the PNR wells. More detailed comments on the failures of the EA at Preston New Road are presented in the Appendix A: Approval of PNR-2 hydraulic fracture plan.

In summary, the overall attitude of the EA, by allowing a mendacious and incompetent interpretation by Cuadrilla to slip through its regulatory net, is reprehensible.

#### 5.12.10. Seismicity at Preston New Road

The OGA commissioned a geophysical consultancy, Outer Limits Geophysics LLC [131], directed by two academic researchers at Bristol University, to carry out an analysis of the microseismicity at Preston New Road triggered by fracking of PNR-1z. Their report for the OGA [132] was published at around the same time in abbreviated form in a peer-reviewed journal [128]. The paper claimed that forecasting larger seismic event magnitudes using microseismic activity statistics could be more effective than relying on a traffic light system. This optimistic conclusion was arrived at before fracking of PNR-2 started. However, the Bristol statistical forecasting methods evidently do not work, because the fracking immediately triggered significant seismicity. The OGA was obliged to suspend fracking 11 days later, on 26 August 2019, after the largest event to date at PNR, M_L_ = 2.9, occurred.

### 5.13. Cheshire: Ellesmere Port

The well was originally planned to explore for coalbed methane within the Coal Measures. It may have been considered that the seismic data available were adequate for this purpose, which was to drill to a minimum depth of 900 m. IGas argued [133] that, since the permit was granted for this minimum depth, it automatically had permission to drill to more than twice that depth (total depth was 1950 m MD; see Appendix A: IGas letter 2017 to MPA Ellesmere Port-1 well). However, this argument is specious; the Nexen planning statement of 14 October 2009, which formed the basis of the permit subsequently granted, states (para. 9.3.6): 

“During the Appraisal Drilling phase two appraisal boreholes will be drilled to an estimated minimum depth of circa 900METRES (3000’) Total Vertical Depth (TVD). The borehole will typically decrease in diameter from 500 mm (20 inches) at the top section to 152 mm (6 inches) in diameter *at the maximum depth in the coal seam*” (emphasis added).

Thus the coal seam target was to be the maximum depth, at whatever depth that might be, and deeper than 900 m. The top of the Pennine Coal Measures came in at 867 m TVD, and two coal seams were drilled at around 1010 and 1040 m MD. Once the drill bit left the Coal Measures and penetrated the Millstone Grit Group at 1,141 m TVD, the drilling was unauthorised. There is no possibility that coal could have been encountered at 1141 m or greater depths. 

The minimum depth of 900 m quoted above probably originates in a condition in the PEDL award. It is customary for the award of a PEDL, if the drilling of a well has been offered by the applicant, to put a minimum depth on the well. This is to avoid the possibility of a licensee later claiming that it had fulfilled the conditions of the award by subsequently drilling a well to shallow depth. The specified minimum depth of c. 900 m corresponds to the likely shallowest depth of the target formation described in the licence application. The licence, PEDL184, started on 1 July 2008. It was awarded for coal bed methane extraction. It is likely that the geological rationale for the siting of the well was optimised for coal bed methane exploration, and it is the later switch of emphasis to prospecting for the deeper shales that has led to problems of interpretation.

The Pentre Chert target layer was initially claimed to be conventional, but IGas later conceded that it was unconventional, and would require acidising techniques to release any hydrocarbons.

In conclusion, a well to target the shales, and in particular, the Pentre Chert, should probably have been sited elsewhere within the PEDL.

Cheshire West and Chester Council, the MPA, commendably refused planning permission on the ground of the non-sustainable nature of the project, stating [134] that:

“The proposal fails to mitigate and adapt to the effects of climate change, ensuring development makes the best use of opportunities for renewable energy use and generation”.

The other regulators played no part in the decision.

### 5.14. DECC/OGA Selective Licence Offers in SE England

DECC (predecessor of the OGA) omitted from the 14th round offer a large swathe of eastern England (Figure 7a). There is nothing about the geology of this region that particularly warrants its exclusion. DECC normally took a pragmatic approach to licensing, and one of the guidelines to licensing new areas is to take into account whether there has been past interest by the oil industry. Evidence for this is best shown by the drilling history. For example, around Witney there are a score or more of old oil exploration wells within a distance of 10–20 km (Figure 7b). Within the east of England zone excluded from the offer there are two ‘islands’ of blocks on offer, labelled A and B in Figure 7a, and although the historical well density is fairly low, there has clearly been past oil industry interest in the region. The Witney constituency itself is densely covered by seismic profiles—another indicator of exploration interest – and the area was licensed for oil exploration in previous awards in 1971 and 1981.

One of the seven UK regional seismic profiles compiled on behalf of DECC by the UKOGL (UKOGL-RG-004) runs north-south through Witney town. Another such profile (UKOGL-RG-005) runs east-west through Lincolnshire, approximately along the upper edge of the map of Figure 7a. That part of Lincolnshire was offered for 14th round licensing, and comparison of the two interpreted regional profiles, which can be freely downloaded, shows that there are no significant differences in the geology of eastern Lincolnshire compared with that at Witney.

Did David Cameron, UK prime minister 2010–2016, instruct DECC, the predecessor of the OGA, to omit his Witney constituency from the areas offered in the 14th round? Figure 7b shows a map of his constituency in blue, with adjacent constituencies outlined in black. The red boundaries with the hatched areas mark the 14th round offer acreage on offer. The acreage is made up of 10 km × 10 km blocks, based on the Ordnance Survey grid. The ‘island’ square labelled B to the east of Witney comprises nine such blocks. Existing onshore oil and gas exploration wells are shown by the red dots.

An FOI enquiry to DECC about the reason for the east of England exclusion zone elicited the following response:

“… the areas included for offer in the 14th Onshore Round were primarily determined by the underlying geology indicating to DECC that hydrocarbons could be present in those regions. However, where an active interest has been expressed by third parties that they would like to explore additional areas for hydrocarbon prospectivity, such areas may be included in the acreage on offer. This includes the 30 km × 30 km region encompassing Brackley, Buckingham and Bicester”.[the area B in Figure 7]

Although this explanation might possibly account for the inclusion of area B within the exclusion zone, it fails to explain adequately why the Witney area (a 30 × 40 km^2^ set of twelve blocks) had been excluded. If area B referred to by DECC and included in the offer was the subject of “active interest” then a more rational 14th round offer map, based on exploration potential, together with expressions of interest, would have been to place the western boundary of the exclusion zone to the east of block B, in the Bletchley–Aylesbury area.

Therefore non-geological reasons must have played a part in the very specific exclusion of the Witney constituency from the 14th round offer. Was there an expression of *disinterest*, and if so, from whom? The third parties referred to by DECC who were allegedly interested in block B may or may not have subsequently applied for the blocks on offer, but in either case no award was made. This outcome calls into question the veracity of DECC’s explanation as quoted above.

### 5.15. Non-Fulfilment of Licence Commitments

INEOS provides an example of licence commitments that cannot realistically be fulfilled. The company holds 29 PEDLs and one exploration licence (EXL273). Three of the licences ostensibly concern CBM (but see above for the two Scottish licences PEDL133 and PEDL162). Excluding the CBM licences and the old EXL273 licence, INEOS has committed to the following work total programme within the first term of the licences, that is, a five-year period starting on 21 July 2016, as shown in Table 1.

This is an extraordinarily ambitious programme, which has no chance of being fulfilled. To date INEOS has acquired only the East Midlands-17 3D seismic survey, of 231 km^2^, between Worksop and Warsop, in 2017; that is, 17% of INEOS’s total 3D commitment. Such action—acquiring 3D seismic before drilling—is laudable, but none of the 21 firm-commitment wells have yet been drilled, nor have any of the 1148 linear kilometres of promised 2D seismic yet been acquired.

To put it another way, INEOS has undertaken to carry out 575 km^2^ of 3D seismic in the East Midlands, where 1144 km^2^ of 3D has been shot since 1996; this would be an addition of 50% to the existing coverage in just 5 years. In North Yorkshire it is promising to carry out 453 km^2^, which will be an addition of 82% to the existing coverage since 1994. In Lancashire and Cheshire the commitment of 410 km^2^ is an addition of 166% to the existing coverage obtained since 1998. In summary, INEOS committed in 2017 to undertake within five years some 1363 km^2^ of 3D, in areas where the entire UK onshore hydrocarbon industry had only achieved 1969 km^2^ in the previous 20 years—this target is logistically unrealistic, even if finance is not a problem.

The sole UK-based seismic contractor in the last decade, Tesla Exploration, went into receivership in 2016, and the company was dissolved [135] on 2 July 2020. Its assets had been bought by IMC GSL Limited in 2016, but it is not clear whether this new company has any ongoing contracts in the UK. So for INEOS to fulfil its commitments it will probably have to mobilise contractors from mainland Europe. To plan, organise, mobilise, acquire, process, and interpret a 3D survey requires at least one year, probably two, especially if the contractor has to come from mainland Europe.

Similarly, there are appears to be an insufficient number of UK-based drilling companies with equipment and personnel capable of penetrating to 4000 or 4500 m, the total depth of most of the INEOS commitments, for the company to fulfil its drilling and horizontal fracking promises. The initial terms of most of the PEDLs expire in July 2021, and hardly any of the promised seismic and drilling has yet been undertaken.

In my view this non-commitment problem rests not so much with INEOS as with the OGA, for granting all 29 licences to INEOS with conditions (Initial Term work programmes) that will be difficult or impossible to fulfil. The OGA must have known in 2016 that no one operator could possibly fulfil such obligations. Therefore it may be concluded that the OGA knew in advance that numerous extensions would need to be granted.

Drill or Drop reported [136] on 20 May 2020 that extensions to the licence term have been sought and granted by the OGA for 20 licences awarded in 2016. One licence dating back to 2008 (Europa, PEDL143 at Leith Hill, Surrey) has been extended six times despite no substantive work having been carried out [137].

The OGA attitude to licence commitments, with extensions freely granted, makes a mockery of the PEDL system.

## 6. Synthesis

### 6.1. Summary of Case Histories

Table 2 summarises the case histories concerning specific sites and how they have been regulated. Table 3 provides the key to summary comments encoded in Table 1.

### 6.2. Methods Used by the Companies

#### 6.2.1. Misleading Geological Information Provided by the Operators

Planning applications may be salami-sliced in two ways—temporal and/or spatial. Temporal salami-slicing is when an application is made for conventional drilling when it is clear from the target that unconventional methods will be required later for development and production. This is also known as ‘gateway’ drilling. Areal salami-slicing involves submitting one project as a series of separate applications, each ostensibly independent.

Unconventional exploitation has been reclassified as conventional. The methods proposed by UKOG and other operators in the Weald to exploit the unconventional resource of the Kimmeridge Clay Formation have been progressively watered down over a decade, presumably in an attempt to deflect criticism of the public over its justifiable fear of fracking. Part of this softening up process has been to avoid the use of the word ‘shale’, and, instead, to mendaciously refer to the micrites as ‘limestones’. They are merely thin (<30 m) layers of calcareous mudstone. They are only half-way, at best, to being limestones (i.e., pure CaCO_3_); nevertheless, they are routinely referred to as ‘limestones’ by the operators in the Weald (see Appendix A: Micrites in the Kimmeridge Clay Formation). This is an attempt to mislead the MPAs and the public, by avoiding the use of words like shale or mudstone. One or more micrite layers have been targeted by horizontal or oblique drilling, but since the natural fractures in the micrite are vertical, any oil flow will tend to come from above or below the wellbore, i.e., from the shale.

A similarly misleading geological term, used by IGas at Ellesmere Port, is chert, which is simply a siliceous shale.

Geological information presented to the regulators may be misleading or even mendacious; the use by IGas at Ellesmere Port of a seismic line some 8 km to the east of the well in question as a well-tie. A well-tie (of seismic to well data) is normally and correctly done with a seismic line passing through the well or in its vicinity.

The claims made by UKOG about Horse Hill-1 proving a ‘world-class’ oil discovery are misleading. On several occasions and at several sites Cuadrilla provided misleading press releases about its drilling problems, which it was trying to hide. However, MPA councillors and officials may be influenced by such misleading reports to support developments of which they might otherwise have been more critical.

Dart Energy at Airth presented a cross-section with important faults removed.

Angus drilled an unauthorised sidetrack at Brockham; IGas drilled about twice as deep as was permitted at Ellesmere Port; only the first of these transgressions was picked up by the MPA. However, the MPA was eventually obliged to back down to avoid a potentially costly legal battle. UKOG at Broadford Bridge drilled to a different target from that permitted by the MPA.

Celtique Energie’s example seismic line for both Fernhurst and Wisborough Green was misleading, because it was nowhere near either well site. The location of this line was not disclosed by Celtique, but is shown by the green wavy line in Figure 3b, which I identified after a search of the UKOGL database. Fernhurst is about 2 km west of the west end of this line, and Wisborough Green about 7 km east of the east end. The nearest seismic lines to Fernhurst lie 500 m to the north, and the nearest to Wisborough Green is about 100 m to the north. At both sites the proposed horizontal wells would be towards the SW, passing nowhere near any existing seismic lines.

The green wavy line shown in Figure 3b is TER-91-06, for which the images and side label are available to view on the UKOGL website. The version shown by Celtique has been reprocessed, and depicts nine colour-coded interpreted horizons. The structure appears to be flat, and no faults are visible. However, the original version, which has been time-migrated, shows clear evidence of faulting. Comparison of the old and the new versions of the line suggest that the reprocessing has smeared out fine details of the structure. This can be achieved, for example, by excessive use of residual statics. The reprocessed version therefore gives a misleading picture of unfaulted geology. Celtique also submitted a misleading diagram in each application purporting to show that the target limestone was a conventional oil trap, despite being located in the axis of a regional syncline. Lastly, Celtique portrayed a 9 km long north–south shallow geology cross-section, with the Fernhurst proposed wellsite mispositioned by 400 m.

None of the above examples of misleading geology were picked up or questioned by the regulators.

#### 6.2.2. Other Methods Used by the Operators to Mislead Regulators

Tactics used by the operators, discussed above, include:Limiting the scope of work (e.g., no fracking in the current proposal, but it might prove necessary in the future).Conditional development.Requesting an excessive time for flow testing as a cover for production (e.g., Angus Energy at Balcombe).Requesting a variation of a permit (e.g., Cuadrilla’s microseismic monitoring at PNR).Denial or concealment of links with academics.Obtaining extensions to the licence term.

Diagrams presented by applicants for an EA permit and/or for MPA planning approval have evolved over the last decade towards providing less quantifiable and verifiable information. At the beginning of the decade the geological information usually included:A well prognosis (a prediction of the layers to be encountered in the proposed well).A structural contour map of the geological layers of interest.An interpreted seismic line through the wellsite.An interpreted geological cross-section through the well.

Number 4 above can substitute for number 3, but is less reliable since no data are presented. There has been a pronounced trend between 2004 and 2019 towards providing fewer and fewer geological data for the MPA and/or the EA to assess. For example, in 2004 Europa at Leith Hill provided items 1–3, following in the tradition of conventional hydrocarbon applications. However, by 2019, in the most recent of my 14 examples, UKOG provided no substantive detail matching items 1–4 above, merely inadequate perspective cartoons purporting to show where the directional wells would go. Furthermore, these cartoons suffer from internal inconsistencies, proving that UKOG has a poor understanding of the geology. It appears that the developers have learned that the less information provided, the better. The regulators have failed to demand the extra information.

Some operators have developed links with academic researchers, but the links remain opaque. For example, Professor Paul Younger and his colleague Dr Rob Westaway of the University of Glasgow wrote a critique of my submissions to Lancashire County Council at the latter’s request in December 2014. It is not clear why LCC asked this particular pair of researchers, out of the dozens of potentially able and willing experts, to undertake this review. An FOI request later proved that they had been in regular contact with Cuadrilla since 2014, and in June 2015 two Cuadrilla staff flew to Glasgow for a meeting with the Glasgow staff. It is therefore possible that Cuadrilla put forward their names to LCC.

The University of Bristol has had ongoing links with Cuadrilla going back to at least 2013. Cuadrilla was formerly a sponsor of Bristol’s BUMPS microseismic research project [138]. However, this information has since been erased from the BUMPS website, although the names of other former sponsors remain. On 31 July 2019 university geophysicists published a joint paper [128] with Cuadrilla about the microseismicity at Preston New Road. The microseismicity dataset had been kept confidential until the OGA released it. A journalist for the *I* newspaper revealed, using emails between Bristol and Cuadrilla [139], that the Bristol researchers and Cuadrilla had agreed in mid-June 2019 to “keep a low profile” about the results of the work. The same Bristol researchers also concealed the fact that the OGA had commissioned, and presumably funded, the research leading to the published academic paper. However, the only sources of funding acknowledged in this paper are three Natural Environment Research Council grants. The implication is that the Bristol authors are concealing a conflict of interest, which is considered serious malpractice in the academic publishing sphere.

The two university-industry links cited above may just be the tip of the iceberg. The problem is not that academia and industry should not collaborate with each other, but that the links are kept hidden. This may lead to deception of the regulators, as in the case of the Lancashire–Glasgow link, as well as evident conflicts of interest; the Bristol research group cited above, for example, can hardly declare itself as independent when it obtained exclusive and confidential access to the microseismic data [140] some six months in advance of its release by the OGA. The OGA could have released the data at the end of 2018.

### 6.3. Which Regulator Has Responsibility for Geological Matters?

The OGA, having awarded a PEDL, avoids taking further interest in geological matters such as the validity or otherwise of the operator’s geological interpretation. However, an exception to this policy is when seismicity occurs. In addition, if a hydraulic fracture plan is required (to be approved by the EA) the OGA will scrutinise that. So, apart from seismicity and hydraulic fracturing, the OGA’s policy is to leave geology to the EA.

The EA’s responsibility concerns the risk of pollution of aquifers from both conventional and unconventional hydrocarbon activity. Clearly this involves an understanding not just of hydrogeology, but also of the deeper geology (1–3 km) where hydrocarbons occur.

The HSE is involved in approving the well design, appointing an independent well examiner to examine:

“information, including information on the design and construction of the well and the sub-surface environment, including the geological strata and formations, the fluids within them and any hazards which the strata and formations may contain”.[141]

Such design depends upon the operator’s interpretation of the anticipated geology, together with any technical difficulties that may be foreseen for example, when drilling at an inclined angle, and/or through a fault. It is pertinent to note that the HSE seems to depend upon industry sources for its own regulations; for example, an HSE presentation [142] refers to the Oil & Gas UK (OGUK) guidelines for well life cycle integrity, and for well abandonment; and to UK Onshore Oil and Gas (UKOOG) for shale gas well guidelines, rather than using its own. OGUK and UKOOG are industry bodies, in this case setting the standards for the regulator to follow.

The BGS publishes national, regional and local geological interpretations. It gives advice to, and collaborates with, both the EA and the OGA. However, it avoids becoming involved in giving opinions on local geology relevant to an operator’s planning applications.

The planning departments of MPAs do not have their own geological expertise. They depend on the other agencies. As we have seen, the MPA’s role in the planning process requires that the other agencies must be assumed to be doing their job properly. However, an MPA can from time to time call upon earth science experts to give it an opinion on a particular geological aspect of a planning application.

The Broadford Bridge history illustrates how the OGA seems to ignore the crucial details of the MPA permit. Clearly the two regulators do not communicate with each other. Similarly at Canonbie (Section 5.2 above) we saw DECC and the Coal Authority, acting in apparent ignorance of each other’s regulation.

### 6.4. Wherein Lies the Regulatory Geological Expertise in the Scrutiny of Industry?

This question goes to the heart of the subject of this review. My contention is that the depth of required expertise is either lacking, and/or is distributed in such a manner between the various regulators that sound decisions may not be arrived at. Furthermore, whatever the decision, no one agency will take responsibility.

It is unlikely that any of the earth science personnel in any of the regulatory bodies have sufficient experience and knowledge of industry to scrutinise effectively the geological content of planning applications for unconventional exploitation. The regulators, particularly, the OGA and the EA, need the expertise of a few ‘poachers turned gamekeepers’, but such expertise is lacking because the UK unconventional industry is only a decade old.

The expertise in the BGS tends to comprise career earth scientists; there are (or were) numerous examples of such scientists, including myself, who started their career during the phase of intensive commissioning of oil and gas work by the Department of Energy in the 1970s and 1980s. But the flow of expertise is invariably from the BGS into industry, and never the other way. As with the regulators, hands-on expertise in unconventional exploitation is probably absent within the BGS.

The competence of MPAs in scrutinising the geological aspects of planning applications has evolved little in the last decade. Part of the problem is that over the last decade the industry has become adept at submitting less and less geological information of substance, as discussed above.

## 7. Conclusions

The major issue is the potential for permanent contamination of shallow groundwater resources by unconventional extraction techniques, which irreversibly alter the hydrogeological properties of the deep subsurface.

The types of potential pollution which may occur from fracking and its related activities include groundwater and surface water pollution by brine, heavy metals, radioactive elements (both heavy and light) and by extremely toxic chemicals used in fracking such as polyacrylamide; methane and H_2_S emissions; disposal of produced water; and earthquakes. This is a long-term potential problem; even if no significant contamination or damage has been detected to date in the decade since the UK started unconventional exploration, it does not imply that it might not arise in the future; nor does it imply that regulation has been successful to date. 

The case histories demonstrate a lax, fragmented, and frequently incompetent regulatory regime which has no overarching geological remit or understanding, and which seems to be driven by the government’s directives, now expressed through the OGA, to maximise oil and gas production.

The OGA allows licence obligations lie unfulfilled. No sanctions on the licensees are ever applied, and there appears to be no effective means of cancelling the licences of inadequately performing operators. The technical level of Cuadrilla in Lancashire has remained abysmal despite both its competence and integrity having been called into question as long ago as 2012. INEOS has been granted licences with no hope of the obligations ever being fulfilled. These are examples of failure of regulation by the OGA.

Two regulators—DECC (now the OGA) and the Coal Authority (then a subsidiary of DECC)—were both independently licensing the same geological target, the coal seams at Canonbie. This illustrates the lack of communication between regulators.

The operators have learned to game the system by providing progressively less geological information, and by the use of misleading or anodyne geological descriptions, avoiding words like fracking or shale.

The MPAs and their staff do not have any special expertise in geology. A planning decision may therefore be overly dependent on what the developer chooses to disclose about the geology. External geological expertise requested by an MPA has, on occasion, proved to be poor; experts may have been recommended by the developer, thus giving rise to a potential conflict of interest. The links between academic experts on the one side and regulators and industry on the other side have sometimes been suppressed.

Developers frequently breach planning consent in non-trivial ways. However, one MPA challenged the legality of unauthorised drilling at Brockham, but only requested a post facto regularisation of the illegal drilling. The question of the unconventional nature of this drilling was never addressed by regulators.

Cheshire West and Chester Council, the MPA for the Ellesmere Port development, commendably refused planning permission for IGas on the ground of the non-sustainable nature of the project, but it is not known at the time of writing whether this decision will be overturned by the relevant minister of state.

There is a history of inadequate record-keeping by the HSE and EA at Brockham, Horse Hill, and Leith Hill, which works to the advantage of the developers.

The EA and SEPA, upon whom the burden of the geological aspects of regulation falls, have both proven to particularly inadept at addressing major geological inconsistencies and errors. On the rare occasions when the EA has asked for advice from the OGA or from the BGS, the response has been either inadequate or not forthcoming. Several permits were issued by the EA and SEPA in clear disregard of environmental risks, preferring to accept the mendacious or erroneous accounts of the geology supplied by the developer.

Two planning inquiry reports have been called in for determination by the appropriate minister, but after an unduly long period (6 years in the Airth case, 18 months in the Ellesmere Port case) no determination of either of them has been made. Such unreasonable delay is a failure of regulation.

In conclusion, UK regulation of the subsurface aspects of onshore unconventional hydrocarbons is very far from being of gold standard; it is woefully inadequate. Alan Tootill wrote of the regulators in early 2013 [44]:

“They are on a learning curve … If shale gas production really gets going in the UK, the regulators will be learning on the job. This is bad news for the public and for the environment”.

However, the case histories show that little or nothing has been learned by the regulators in the intervening seven years, regarding the subsurface aspects of unconventional hydrocarbon regulation. It would appear that Tootill’s learning curve has flat-lined.

## Figures and Tables

**Figure 1 ijerph-17-06946-f001:**
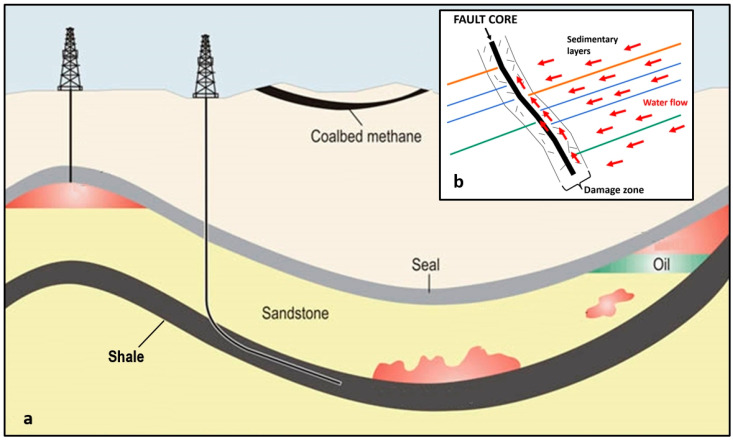
(**a**) Conventional and unconventional oil and gas location in geological structures. This cartoon, from the US Energy Information Administration, shows simple US-style geology with a thin, unfaulted shale layer. (**b**) Cross-section of water flow up a fault zone. The core, central part of a fault (black) may either be transmissive or else be a barrier. The damage zone on either side, comprising fractured rock, will usually be transmissive. The resulting conceptual water flow from right to left is indicated by red arrows.

**Figure 2 ijerph-17-06946-f002:**
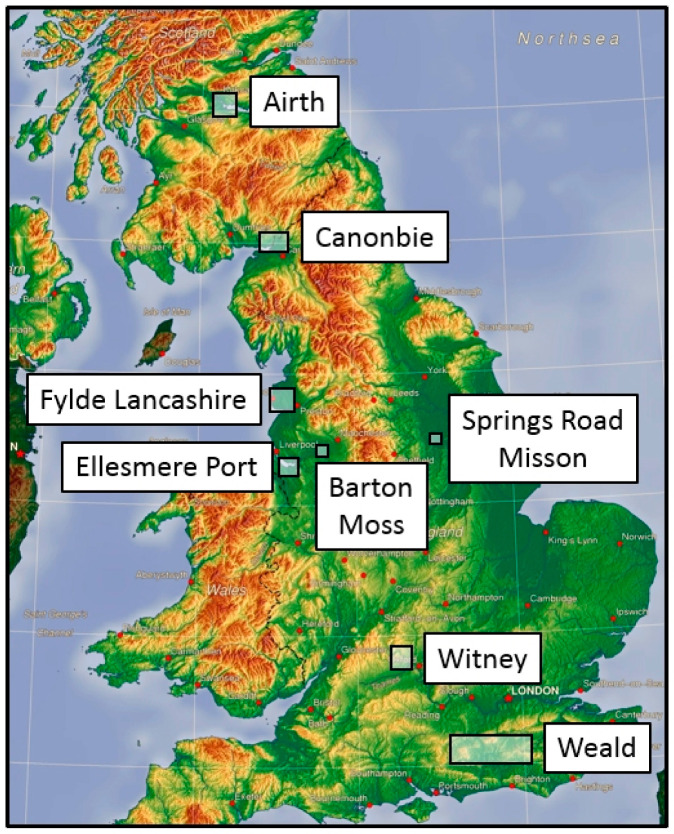
Location map of the case histories discussed in the text.

**Figure 3 ijerph-17-06946-f003:**
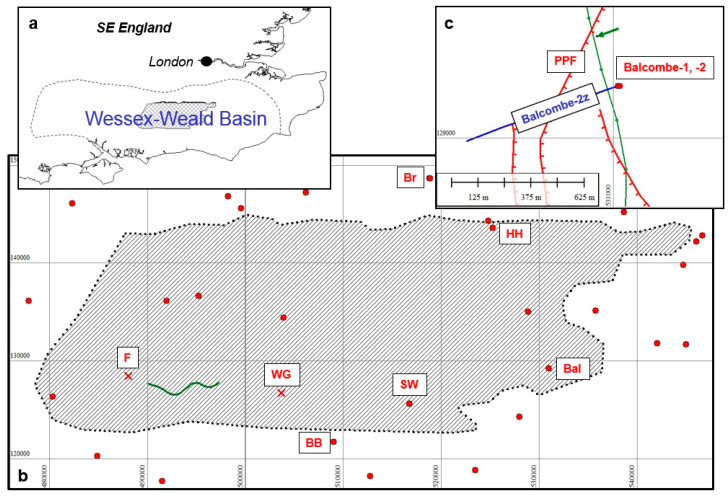
(**a**) Location map of the Wessex–Weald basin. (**b**) Weald Basin with area of mature Kimmeridge Clay Formation shown by hatching. Wells referred to in the text are labelled: Br—Brockham; HH—Horse Hill; SW—Southwater; Bal—Balcombe; BB–Broadford Bridge. Crosses are proposed well locations: F—Fernhurst; WG—Wisborough Green. Wavy green line—2D seismic line pictured by Celtique Energie. (**c**) The Balcombe locality: mapped surface normal faults are shown by red lines with teeth on the downthrown side (PPF—Paddockhurst Park Fault). Blue line—subsurface location of horizontal well Balcombe-2z. Green line—2D seismic line used in planning application by Cuadrilla. Green arrow—location of Balcombe-1 as mispositioned by Cuadrilla.

**Figure 4 ijerph-17-06946-f004:**
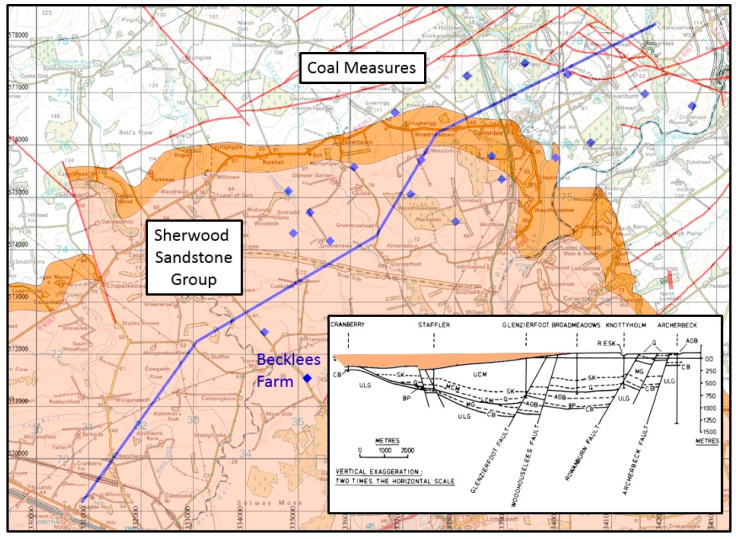
Drill sites at Canonbie (blue diamonds). The unconfined primary aquifer of the Sherwood Sandstone Group is shown in light orange; underlying Permian sandstones in darker orange. Faults are shown by red lines. The blue line is the location of the cross-section of Picken [57] shown in the inset. Contains BGS data © Crown copyright and database right (2020). A BGS/EDINA supplied service.

**Figure 5 ijerph-17-06946-f005:**
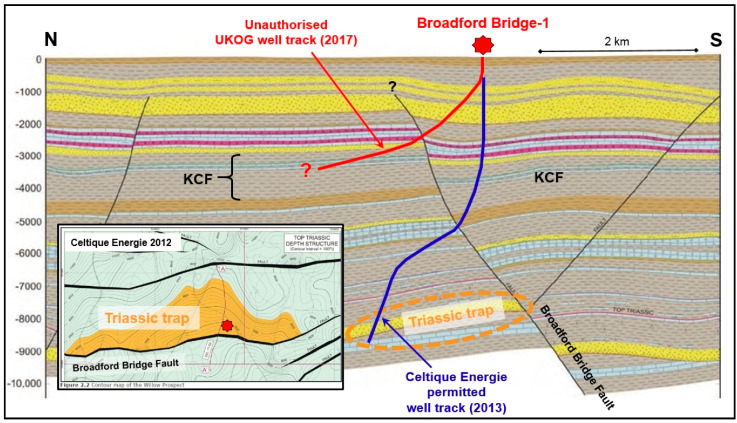
North–south geological cross-section through the Broadford Bridge-1 well drilled by UKOG in 2017. Vertical scale is shown in feet. The MPA-permitted well track of Celtique Energie is shown in blue; the approximate track of the unauthorised well drilled by UKOG in 2017 is shown in red. The inset map shows the structure map of the Triassic trap named ‘Willow’ by Celtique Energie.

**Figure 6 ijerph-17-06946-f006:**
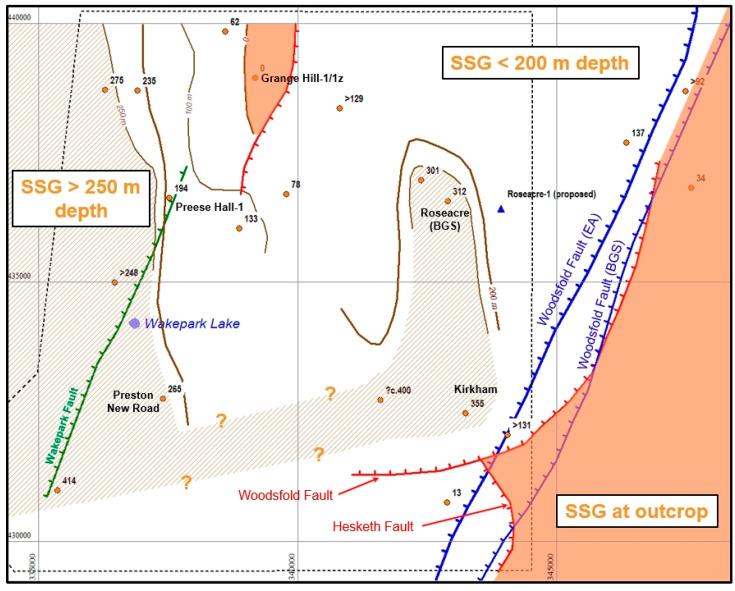
The Fylde, Lancashire. The Sherwood Sandstone Group (SSG) is at outcrop east of the Woodsfold and Hesketh Faults (red) and in a small upfaulted block where Grange Hill-1/1z was drilled. Contours on the top SSG are shown at 0, 100, 200 and 250 m. The cross-hatched area shows where the SSG is greater than 250 m deep. The dashed outline is the area covered by the Bowland-12 3D seismic survey. Two prior versions of the Woodsfold Fault are shown in blue. Roseacre-1 (blue triangle) was Cuadrilla’s proposed well at Roseacre Wood, distinct from the 500 m deep Roseacre monitoring well drilled by the BGS. Numbers attached to boreholes (orange dots) indicate depth in metres below sea level to the top of the SSG.

**Figure 7 ijerph-17-06946-f007:**
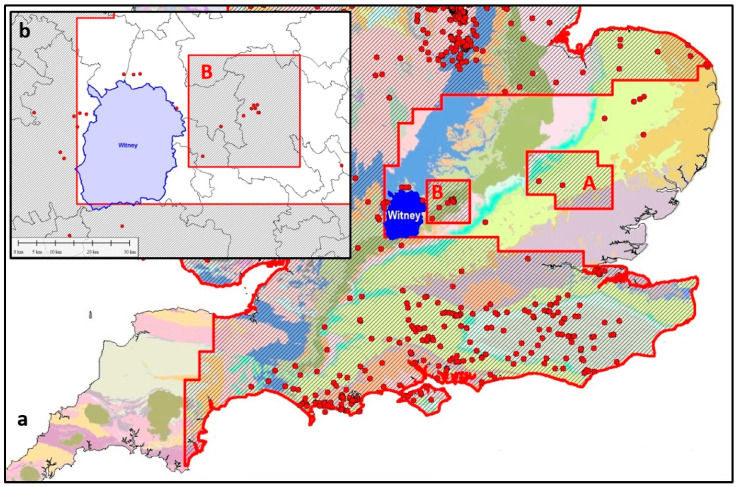
(**a**) Areas offered by the Oil and Gas Authority (OGA) in the 14th onshore licensing round by the OGA (hatched areas with red boundaries) overlain on the regional geological map of SE England. The Witney parliamentary constituency is shown in solid blue. Exploration wells up to 2015 are shown as red dots. (**b**) Detail of the Witney constituency (blue) omitted from the offer areas (hatched), with other parliamentary constituency boundaries outlined in black. Block B was allegedly included in the offer by request, but finally was never awarded. Contains BGS data © Crown copyright and database right (2019). A BGS/EDINA supplied service.

**Figure 8 ijerph-17-06946-f008:**
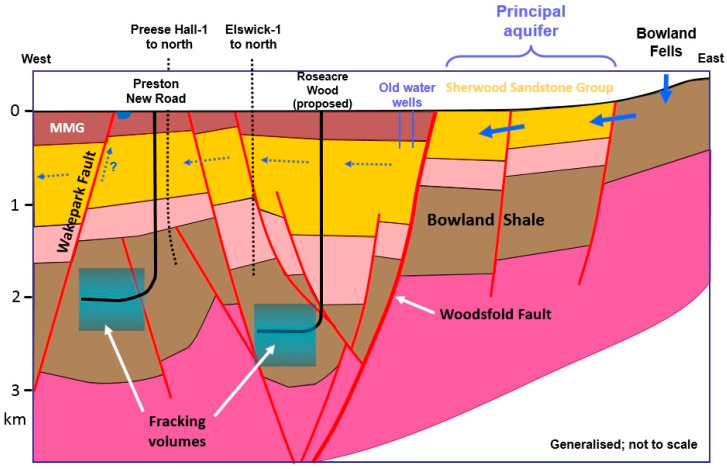
Schematic cross-section across the Fylde, showing the unconfined Sherwood Sandstone Group principal aquifer (SSG; yellow) east of the Woodsfold Fault, but confined below the Mercia Mudstone Group (MMG; rust-brown) west of the fault. The Bowland-Hodder Group (shale) is shown in light brown, the pink layer is the combined Manchester Marls (upper) and Collyhurst Sandstone (lower). Roseacre Wood was a proposed Cuadrilla fracking site. The Wakepark Lake is indicated by the blue cup shape adjacent to the Wakepark Fault.

**Figure 9 ijerph-17-06946-f009:**
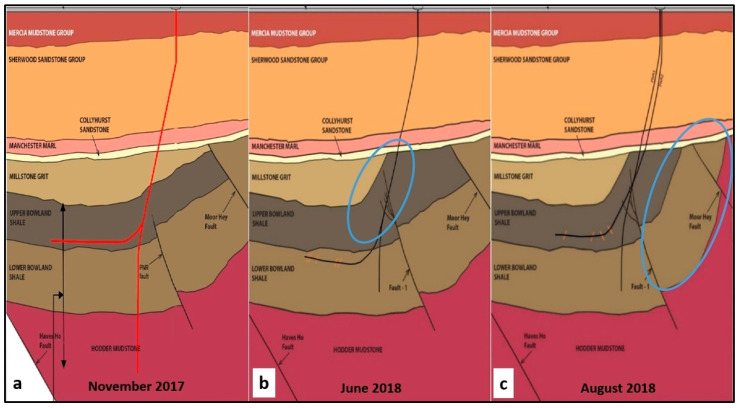
East–west cross-sections through Preston New Road by Cuadrilla in support of its hydraulic fracture plans, showing progressive changes to the geological interpretation. (**a**) November 2017; proposed PNR-1 and 1z (shown in red), prognosed to penetrate 300 m of Millstone Grit (buff). (**b**) June 2018; revised interpretation (blue ellipse) to explain absence of Millstone Grit at the well (black line). (**c**) August 2018; further revision of the geology (blue ellipse) for the PNR-2 HFP, to minimise the unrealistic thickness variations of the Bowland Shale shown in (**b**). Interpretations (**b**) and (**c**) are arrived at by ‘photoshopping’ the original image (**a**), and are not based upon geological re-interpretation of the seismic dataset. The original images have been slightly compressed horizontally.

**Table 1 ijerph-17-06946-t001:** INEOS work commitments to be completed within 5 years from 21 July 2016.

Region	2D Seismic (km)	3D Seismic (km^2^)	Firm Wells	Horizontal Wells, Fracked
East Midlands	550	575	8	4
Lancs/Cheshire	410	335	6	2
North Yorks	188	453	7	5
Totals	1148	1363	21	11

**Table 2 ijerph-17-06946-t002:** Summary of regulation of operating companies.

Company	Well or Site	Technical Failure	Misinformation	OGA	MPA	EA/SEPA	HSE	Notes
Angus	Brockham	A1	A2		A3			
Celtique	Wis. Green/Fernhurst		C1					C2
Cuadrilla	Anna’s Road-1	Cu1	Cu2					
	Grange Hill-1/1z	Cu3	Cu4					
	Balcombe-2/2z	Cu5	Cu6, X					
	Preese Hall-1	Cu7	Cu8			Cu9	Cu10	
	Roseacre Wood	Cu11						Cu12
	Preston New Road	Cu13	Cu14	Cu15		Cu16		
Dart	Canonbie		D1			D2		
	Airth CBM/shale	D3	D4					D5, X, Y
Europa	Leith Hill			E1		E2		E3
IGas	Ellesmere Port-1		I1-3, X		I4			Y
	Tinker Lane	I5						
	Springs Road	I6	I7					
	Barton Moss		I8	I9	I9			I10
INEOS	29 PEDLs	IN1		IN1				IN2
UKOG	Broadford Bridge	U1	U2, X	U3	U4	U5		U6
	Horse Hill	U7	U8-9, X	U10	U11		U12	

**Table 3 ijerph-17-06946-t003:** Key to Table 2.

OGA	Oil and Gas Authority
MPA	Mineral Planning Authority
EA/SEPA	Environment Agency/Scottish Environmental Protection Agency
HSE	Health and Safety Executive
X	Unconventional development disguised as conventional exploration
Y	Public inquiry held
A1	Drilling attributed to mix-up over wellheads at wellsite
A2	Brockham-X4 drilled without permission
A3	Surrey CC allowed Angus to apply for and get retrospective planning approval
C1	Seismic example used section with faulting ‘wiped out’ by reprocessing
C2	Both applications refused; licences sold to UKOG
Cu1	Tool stuck, so well abandoned
Cu2	Abandonment stated to be due to ‘over-wintering birds’
Cu3	Drilling assembly stuck; sidetracked but well abandoned
Cu4	Well allegedly suspended in May 2011 but drilling continued till 19 July 2011.
Cu5	Drilled blind through shallow fault known from BGS maps
Cu6	Application for conventional approved then proposal to frack added
Cu7	Fracking triggered 2.3 and 1.5 Ml quakes; wellbore flattened; well abandoned
Cu8	Minister criticised Cuadrilla’s performance and withholding of information
Cu9	No response to MPA request
Cu10	Failed to visit; no checks after plug & abandonment
Cu11	Poor geological interpretation despite using 3D seismic survey
Cu12	Site proposals abandoned
Cu13	300 m of predicted Millstone Grit found to be missing
Cu14	Attempts to patch up defective interpretation for Hydraulic Fracturing Plan (HFP)
Cu15	Release of 3D seismic survey held back for 10 months
Cu16	HFP for PNR-2 approved in spite of its misleading geological interpretations
D1	Salami slicing of project into separate applications
D2	SEPA approved drilling into principal aquifer with inadequate well safeguards
D3	No plan for avoiding/mitigating faults during horizontal drilling
D4	Major faults omitted from cross-sections through site
D5	Since taken over by IGas then INEOS
E1	Six licence extensions to PEDL143 granted
E2	EA issued permit despite risk of contamination of public water supply boreholes
E3	Well never drilled; licence taken over by UKOG
I1	Well drilled to twice permitted depth
I2	Claimed that target Pentre Chert is conventional
I3	Geology from 8 km to east misleadingly used as site geology
I4	Permission refused on environmental grounds, but IGas appealed
I5	Failed to penetrate expected Bowland Shale, December 2018
I6	Misson Springs Fault through wellsite not identified
I7	Risk to groundwater hidden by misleading geological cross-sections
I8	Permit to drill to coal for CBM but drilled to Bowland Shale
I9	MPA knew of impending breach but passed responsibility to EA.
I10	PEDL193 now held by INEOS
IN1	Impossible to fulfil all its licence obligations in agreed licence timeframe
IN2	First CEO was a chemical engineer, not a reservoir engineer or geologist
U1	Drilled through fault zone, sidetrack but still drilling problems; well suspended
U2	Drilled at different angle and to different target from that permitted by MPA
U3	Repeated granting of extensions
U4	No sanction applied for serious breach of permit
U5	Concerns on change of target and drilling ignored; well spudded before EA permit issued.
U6	Inherited licence and permit to drill vertically from Celtique in 2016
U7	Severe geological errors; nearby old well mispositioned by 150 m; fault zone drilled
U8	Target Kimmeridge so-called ‘limestone’ layers are micrite (calcareous shale)
U9	‘Gatwick Gusher’ claims founded on conflating oil in place with recoverable reserves
U10	Surrey earthquake swarm, summer 2018, concluded to be natural
U11	Ignored consultation concerns on geology, acidisation, earthquakes
U12	Daily logs of wellhead activity relevant to 2018 earthquakes ‘unavailable’

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
