# Peer review of "Inadequate Regulation of the Geological Aspects of Shale Exploitation in the UK"

_ijerph, 2020, doi:10.3390/ijerph17196946_

Round 1

Reviewer 1 Report

This is a comprehensive, extremely well-researched, and definitive study of onshore unconventional O&G development in the UK.  Prof. Smyth shows he has command of the geological aspects of such development that are the bases for his constructive criticism of the administrative/regulatory shortcomings revealed by this paper: he is well-qualified to unleash such criticism.  UK shale is NOT U.S. shale!

This is a timely expose of a confusing industry/government relationship that has repeatedly led to increased risks to the public, with little promised return. Of course, the structure and length of the paper are as unconventional (for a technical journal) as the wells themselves. The unconventional structure of the paper (it is definitely NOT Introduction, Methods, Results, Discussion) is appropriate to the task in this case. I leave it to the journal editors whether shortening should be required: I think it is not.

Reviewer 2 Report

There i a fundamental lack of understanding of the legal framework, see for instance the general statement at 279-282. this is incorrect (eg refer to research done on retention leases in Australian offshore petroleum which encourages land banking) 

358-364 regarding HSE is short, misleading and does not capture the essential role of the hHSE in analysing geology and engineering related to wells and integrity 

Author fails to understand the fundamental concepts of regulation, particularly relating to the principles of regulation. S/he conflates issues regarding work program, discretionary bidding and new technologies. the regulatory framework is such that it addresses new technologies for drilling through the risk analysis undertaken by HSE. If the author understood the role of HSE, this would be clearer. Also, in relation to award of licence, a consideration and understanding of the 1994 Hydrocarbon Directive (894/22/EC) is warranted in relation to how a license is/must be awarded 

No addressing of significant legal issues raised by the Bocardo v Star Energy case and the resultant amendment to the Energy Infrastructure Act. 

Author Response

Reviewer no. 1: lines 279-282: the original text read:
“State regimes controlling their oil and gas resources abhor licence speculation; that is, the acquisition of licences which are then held either without development, akin to land-banking, and/or are traded with other companies. Historically UK system described above prevented such unproductive activity; however, under the OGA this protection has been loosened.”

The reviewer asserts that I have a “fundamental lack of understanding” of the legal framework, but only provides one possible counter-example to what I have written, that is, the lease retention system offshore Australia. No examples from the UK regime are proposed.

The reviewer’s comment about the offshore Australian regime encouraging land banking is misleading. The system permits the conversion of an exploration lease to a retention lease, in which the lease may remain dormant for up to 15 years. But there has to have been a prior discovery which is expected to become commercially viable at some point in the future. That is not land banking. I have added another section, A note on licensing regimes outwith the UK, to the supplementary information, going into more detail on retention leases.

Nevertheless, to dealt with these points I have altered and expanded on the two sentences quoted above (originally lines 279-282) to read (lines 278-287):

“Most state regimes controlling their oil and gas resources abhor licence speculation; that is, the acquisition of licences which are then held either without development, akin to land-banking, and/or are traded with other companies (see SI: A note on licensing regimes outwith the UK). Historically UK system described above prevented such unproductive activity; however, under the OGA this protection has been loosened. Despite the OGA’s assurance that by splitting the licence term into three parts:
“It allows the OGA to ensure that licensees do not retain valuable exclusivity of hydrocarbon exploration and extraction without doing enough work for this to be justified.”i
licences are now awarded when: ….”

Reviewer comment, para. 2

358-364 regarding HSE is short, misleading and does not capture the essential role of the hHSE [sic] in analysing geology and engineering related to wells and integrity

Response, para. 2

The original text reads:
“3.7 The Health and Safety Executive
The Health and Safety Executive (HSE) is a statutory consultee of the MPA. In the context of geology, its remit is to advise on well construction, and to monitor the wellsite during and after drilling and completion. While its geological role is limited relative to the other agencies, there are several examples of inadequate oversight and/or record-keeping by the HSE which have proved to be geologically pertinent.”

The HSE appoints an independent well examiner to examine:

“information, including information on the design and construction of the well and the sub-surface environment, including the geological strata and formations, the fluids within them and any hazards which the strata and formations may contain” [HSE 2017. Well Examination Scheme Inspection Guide. TRIM 2017/367996. Available at https://www.hse.gov.uk/offshore/]

Such examination is evidently limited to the geology (as presented by the operator) to be bored through, and thus cannot properly take into account the wider geological or hydrogeological context. Therefore I stand by my assertion that the geological role of the HSE is limited, relative to that of other regulators.

So I have expanded my discussion of the HSE under section 6.2 (original lines 2018-2020) to read:

“The HSE is involved in approving the well design, appointing an independent well examiner to examine:
“information, including information on the design and construction of the well and the sub-surface environment, including the geological strata and formations, the fluids within them and any hazards which the strata and formations may contain”132
Such design depends upon the operator’s interpretation of the anticipated geology, together with any technical difficulties that may be foreseen for example, when drillig at an inclined angle, and/or through a fault. It is pertinent to note that the HSE seems to depend upon industry sources; for example, an HSE presentationii refers to the Oil & Gas UK (OGUK) guidelines for well life cycle integrity, for well abandonment, and to the UK Onshore Oil and Gas (UKOOG) for shale gas well guidelines, rather than using its own. OGUK and UKOOG are industry bodies.”

Reviewer comment, paras. 3, 4
Author fails to understand the fundamental concepts of regulation, particularly relating to the principles of regulation. S/he conflates issues regarding work program, discretionary bidding and new technologies. the regulatory framework is such that it addresses new technologies for drilling through the risk analysis undertaken by HSE. If the author understood the role of HSE, this would be clearer. Also, in relation to award of licence, a consideration and understanding of the 1994 Hydrocarbon Directive (894/22/EC) [sic: the correct OJ directive is 94/22/EC] is warranted in relation to how a license is/must be awarded
No addressing of significant legal issues raised by the Bocardo v Star Energy case and the resultant amendment to the Energy Infrastructure Act.

Response, paras. 3,4
For comments on the HSE, see above.

On the award of a licence, the reviewer has gone off topic, having forgotten that the scope of the review is limited to geological aspects, as specified by its title. So reference here to the EU directive is irrelevant, since it contains no mention of geology, other than a mention of granting a licence of an area to a licensee of a contiguous area. Similarly, the mention of the trespass case (Bocardo vs. Star Energy) has nothing to do with regulating geology; it was purely about right of access to minerals below someone’s land.

i OGA Consolidated Onshore Guidance, March 2018, p.10. Available at
https://www.ogauthority.co.uk/media/4693/march-2018_consolidated-onshore-guidance-compendium_vfinal.pdf
ii Bradley, P. and Norman, C. 2016. How HSE regulates onshore oil and gas. Presentation to joint US/EU conference on health and safety at work, Bakken safety tour

Reviewer 3 Report

Ijerph - 888434

Inadequate regulation of the geological aspects of shale exploration in the UK

The proposed review deals with an important topic in the UK: the safety of the shale gas exploration and production. This is a valid review and I recommend publication of this manuscript in the International Journal of Environmental Research and Public Health. Despite this, corrections are needed. All the general and specific comments need to be addressed and I am available under request of the editors to review the manuscript a second time.

General comments

- I think the author should make the point that the validity of this review has also implications for production of shale gas in other areas of NW Europe such as France, Netherlands, Denmark and Germany. Indeed, the stratigraphy is similar with the Triassic Sandstone and the Cretaceous Chalk as major aquifers

- The author needs to make clearer that the major hazard is represented by the protection of groundwater resources in the Sherwood Sandstone aquifer and also the Chalk. Seismic hazard might be a minor issue due to the tectonic setting of Great Britain. The authors need to specify which is the magnitude of the earthquakes that fracking can produce in the examined case study. If the author is able to support the opposite that the seismicity is important even in that case more detail is needed

- The authors missed recent papers on the topic that were presented in a specific conference on hydrogeology, contamination risk and production of shale gas at the Burlington House. The organizers were the Geological Society, IAH and the British Geological Survey. Please, refer to the presented papers (see list in the specific comments)

- Some details on the hydrogeology of the Sherwood Sandstone aquifer are not correct or not supported by supporting documents

- Please, follow the minor comments to fix the points that I made above

Specific comments

  1. Introduction

Lines 122-160. You need to make clear from the beginning that the contamination risk in the UK is high for the two major aquifers: the Chalk and Sherwood Sandstone aquifer. This is related to the occurrence in the startigraphy of prospective horizons for shale gas at the bottom of these aquifer-units. See recent paper below of the British Geological Survey:

- Loveless, S.E., Bloomfield, J.P., Ward, R.S., Hart, A.J., Davey, I.R. and Lewis, M.A., 2018. Characterising the vertical separation of shale-gas source rocks and aquifers across England and Wales (UK). Hydrogeology journal26(6), pp.1975-1987.

Lines 44. I think seismicity in the Great Britain is not the main issue on shale gas production. However, the key point is in my opinion the protection of groundwater resources from contamination due to the connectivity between shallow and deep aquifer-systems.

  1. Scrutinity of geological aspects of the licensing issues

Lines 210-215. What about mentioning also Shell and TOTAL active on exploration and production of hydrocarbon resources in the UK?

Lines 231-236. Here, you need supporting references on the hydraulic behaviour of faults in sedimentary basins. See below

- Bense, V.F., Gleeson, T., Loveless, S.E., Bour, O. and Scibek, J., 2013. Fault zone hydrogeology. Earth-Science Reviews127, pp.171-192.

Lines 248-249. Please, refer to the following recent papers to highlight the conduit behaviour of Cambrian extensional faults

- Medici, G., West, L.J. and Mountney, N.P., 2018. Characterization of a fluvial aquifer at a range of depths and scales: the Triassic St Bees Sandstone Formation, Cumbria, UK. Hydrogeology journal26(2), pp.565-591.

- Medici, G., West, L.J., Mountney, N.P. and Welch, M., 2019a. Permeability of rock discontinuities and faults in the Triassic Sherwood Sandstone Group (UK): insights for management of fluvio-aeolian aquifers worldwide. Hydrogeology Journal27(8), pp.2835-2855.

Lines 294-198. Please, more detail on induced seismicity in terms of the sismo-tectonic background of Great Britain, magnitude of the events and capacity of the faults

Line 346. The majority of the work volume at the EA is on surface-water and shallow (<100 mBGL) groundwater. As highlighted by the author, there is need of geological and hydro-geological experts  on the deep (0.1-2 km) subsurface.

Lines 383-411. You need to insert somewhere in the manuscript information on the magnitude of earthquakes

4 Case histories

Lines 858-866 This is not my field. But, I think the author needs to make clear in the paper that there are two different cases related to extraction of fluids and earthquakes

- Induced seismicity on tectonically active faults. In this case, fracking re-activate faults alternating the cyclicity of the active tectonic structure. Not easy in some case to establish a connection between earthquake and the activity of the fault in this scenario

- Activate a pre-existing palaeo-fault. For example the bounding fault of a basin with Meso-Cenozoic activity in the case of the UK sedimentary basins

I’m asking to the author to better define the seismic hazard related to the shale-gas production in Great Britain.

  1. Discussion

Lines 1097-1103. More geological details to clarify concepts in this part.

Line 1436. Better “damage zone” here

Line 1518-1523. No supporting references here. Please, refer to the recent hydro-stratigraphical schemes from these authors

- Wilson, M.P., Worrall, F., Davies, R.J. and Hart, A., 2017. Shallow aquifer vulnerability from subsurface fluid injection at a proposed shale gas hydraulic fracturing site. Water Resources Research53(11), pp.9922-9940.

- Medici, G., West, L.J. and Mountney, N.P., 2019b. Sedimentary flow heterogeneities in the Triassic UK Sherwood Sandstone Group: Insights for hydrocarbon exploration. Geological Journal54(3), pp.1361-1378.

Lines 1540-1544. Where the SSG is confined the view of the Environmental Agency might be valid. Which are the values of electric conductivity of the groundwater? These values are diagnostic to understand if we’re dealing with a relatively freshwater or not

Line 1546. Some faults in the Sherwood Sandstone aquifer are transmissive other not depending on development of deformation bands vs. open fracture. Generally, the transmissivity is higher or similar to that one of the host rock in the Sherwood Sandstone aquifer. See relevant papers below

- Wilson, M.P., Worrall, F., Davies, R.J. and Hart, A., 2017. Shallow aquifer vulnerability from subsurface fluid injection at a proposed shale gas hydraulic fracturing site. Water Resources Research53(11), pp.9922-9940

- Medici, G., West, L.J. and Mountney, N.P., 2018. Characterization of a fluvial aquifer at a range of depths and scales: the Triassic St Bees Sandstone Formation, Cumbria, UK. Hydrogeology journal26(2), pp.565-591

Also, there are pumping tests available on the Woodsfold Fault?

Line 1579-1583. I have doubt that the groundwater is fresh and drinkable in the Sherwood Sandstone Group at depths > 100-150 mBGL in confined settings. See those relevant documents

- Allen, D.J., Brewerton, L.J., Coleby, L.M., Gibbs, B.R., Lewis, M.A., MacDonald, A.M., Wagstaff, S.J. and Williams, A.T., 1997. The physical properties of major aquifers in England and Wales.

- Medici, G., West, L.J. and Mountney, N.P., 2018. Characterization of a fluvial aquifer at a range of depths and scales: the Triassic St Bees Sandstone Formation, Cumbria, UK. Hydrogeology journal26(2), pp.565-591

Lines 1631-1641. Please, discuss the issue of induced seismicity in connection with the sismo-tectonic background of Great Britain.

Line 1978. “Developers have developed”. Please, avoid repetition

  1. Conclusions

More detail on the risk of contamination of shallow groundwater resources with regards to fracking activity that I think is the major issue

Figures and tables

Fig. 1b Insert the word “core” to the tectonic structure in black in the figure as well as in the text when relevant

References

Please, refer to the relevant papers below. Most of them presented to the most recent conference organized on this topic in the UK (Burlington House, London, summer 2018)

- Allen, D.J., Brewerton, L.J., Coleby, L.M., Gibbs, B.R., Lewis, M.A., MacDonald, A.M., Wagstaff, S.J. and Williams, A.T., 1997. The physical properties of major aquifers in England and Wales.

- Bense, V.F., Gleeson, T., Loveless, S.E., Bour, O. and Scibek, J., 2013. Fault zone hydrogeology. Earth-Science Reviews127, pp.171-192.

- Loveless, S.E., Bloomfield, J.P., Ward, R.S., Hart, A.J., Davey, I.R. and Lewis, M.A., 2018. Characterising the vertical separation of shale-gas source rocks and aquifers across England and Wales (UK). Hydrogeology journal26(6), pp.1975-1987.

- Wilson, M.P., Worrall, F., Davies, R.J. and Hart, A., 2017. Shallow aquifer vulnerability from subsurface fluid injection at a proposed shale gas hydraulic fracturing site. Water Resources Research53(11), pp.9922-9940.
